# Beyond Content Relevance: Evaluating Instruction Following in Retrieval Models

**Jianqun Zhou**[1]*, **Yuanlei Zheng**[4]‡ **Wei Chen**[4]*
**Qianqian Zheng**[1]**, Hui Su**[2]**, Wei Zhang**[1]**, Rui Meng**[3]†**, Xiaoyu Shen**[1,5]†

[1]Ningbo Key Laboratory of Spatial Intelligence and Digital Derivative, Institute of Digital Twin, Eastern Institute of Technology, Ningbo   [2]Meituan Inc.   [3]Salesforce Research
[4]School of Software Engineering, Huazhong University of Science and Technology
[5]Engineering Research Center of Chiplet Design and Manufacturing of Zhejiang Province
`ruimeng@salesforce.com, xyshen@eitech.edu.cn`

## Abstract

Instruction-following capabilities in LLMs have progressed significantly, enabling more complex user interactions through detailed prompts. However, retrieval systems have not matched these advances, most of them still relies on traditional lexical and semantic matching techniques that fail to fully capture user intent. Recent efforts have introduced instruction-aware retrieval models, but these primarily focus on intrinsic content relevance, which neglects the importance of customized preferences for broader document-level attributes. This study evaluates the instruction-following capabilities of various retrieval models beyond content relevance, including LLM-based dense retrieval and reranking models. We develop ***InfoSearch***, a novel retrieval evaluation benchmark spanning six document-level attributes: *Audience*, *Keyword*, *Format*, *Language*, *Length*, and *Source*, and introduce novel metrics – Strict Instruction Compliance Ratio (SICR) and Weighted Instruction Sensitivity Evaluation (WISE) to accurately assess the models' responsiveness to instructions. Our findings indicate that although fine-tuning models on instruction-aware retrieval datasets and increasing model size enhance performance, most models still fall short of instruction compliance.[1]

## 1 Introduction

The advent of instruction-following in large language models (LLMs) has greatly expanded their generative capabilities (Brown, 2020; Lou et al., 2023), allowing users to express more complex intentions through detailed instructions (Black et al., 2022; Touvron et al., 2023). However, retrieval systems have not kept pace with these advancements, continuing to rely on traditional lexical or semantic matching techniques (Xiong et al., 2020; Wang et al., 2022; Xiao et al., 2023). As a result, while users have grown accustomed to interacting with generative models using intricate instructions (Team et al., 2023; Bai et al., 2023; Achiam et al., 2023), their retrieval behavior remains limited to keyword-based queries followed by manual filtering of results to find the desired information. Several studies have started to explore instruction-aware retrievers that can interact with users as seamlessly as generative models, but these primarily focus on task-level instructions (Asai et al., 2023; Wang et al., 2023; Peng et al., 2024), guiding retrievers with one instruction for each task. While this task-level instruction is essential for adapting a single retrieval model to multiple predefined scenarios, it falls short of meeting users' customized demands beyond standard tasks (Weller et al., 2024b).

Recent works have shifted from task-level to instance-level instructions, providing tailored instructions for each instance to better align with customized needs (Weller et al., 2024a; Oh et al., 2024). These approaches specify instructions that which content to include or exclude, thus clarifying user intent. While they greatly enrich the diversity of instructions, their primary focus remains on *content relevance*. When searching for certain documents, users typically care about two aspects: the informational content and its presentation (Taylor, 1962; Mizzaro, 1998), including document-level

---

*Authors contributed equally. † Corresponding authors.
[1]We release our dataset and code on `https://github.com/EIT-NLP/InfoSearch`

attributes such as length, language, and format. A sole focus on content relevance neglects the importance of customized preferences for broader document-level attributes.

We believe that a truly instruction-aware retrieval system must go beyond content relevance to accommodate a variety of user-defined document attributes. To further research in this direction, we propose ***InfoSearch***, a novel benchmark designed to evaluate IR models based on their ability to follow customized instructions across six structured dimensions: Audience, Keyword, Format, Language, Length, and Source. These dimensions encompass key document-level features that address user needs beyond content relevance. Additionally, we include both instructed and reverse-instructed modes to assess the model's ability to comprehend instructions in both affirmative and negative formats. Each instruction is carefully crafted and manually validated to ensure naturalness and representativeness of complex real-world scenarios.

Beyond the comprehensiveness of datasets, well-defined evaluation metrics are essential to thoroughly assess the instruction-following capabilities of retrieval models. While traditional IR metrics like nDCG and MRR are primarily effective for assessing content relevance in ad-hoc retrieval tasks (Weller et al., 2024a), we propose new metrics specifically designed to measure the depth of instruction-following capabilities in retrieval models. By structuring the evaluation across these separate dimensions and modes, we offer a detailed analysis of how well models follow instructions on each condition, providing clearer insights into their strengths and limitations. Overall, fine-tuning on instruction-aware retrieval datasets and increasing model parameters improve instruction-following capabilities, with re-ranking models outperforming retrieval models in this aspect. However, both approaches still show considerable room for improvement in meeting the standards set by our benchmark.

Our contributions can be summarized as follows:

- We propose ***InfoSearch***, an evaluation framework that covers six key dimensions: Audience, Keyword, Format, Language, Length, and Source, to assess retrieval models' ability to follow complex instructions beyond content matching.
- We introduce two novel metrics – Strict Instruction Compliance Ratio (SICR) and Weighted Instruction Sensitivity Evaluation (WISE) – that provide a more nuanced and accurate assessment of retrieval models' instruction adherence compared to traditional IR metrics.
- We evaluate 15 retrieval models, encompassing 1 sparse retrieval model, 8 bi-encoder-based dense models, and 6 LLM-based reranking models. This thorough evaluation enables a comprehensive comparison across diverse methodologies, delivering valuable insights into their instruction-following effectiveness.

## 2 INFOSEARCH: CONSTRUCTION AND EVALUATION

We construct a benchmark, **In**struction-**Fo**llowing **Search** (***InfoSearch***), to evaluating the search models' ability to follow instructions. InfoSearch is composed of query-doc pairs across six dimensions and two novel metrics to measure models' responsiveness to instructions. In this section, §2.1 details the dimension settings and retrieval modes under the InfoSerach framework, §2.2 explains the dataset construction process, and §2.3 describes the design logic behind our proposed metrics.

### 2.1 DATASET FRAMEWORK

In real-world scenarios, users exhibit a wide range of complex and diverse search intentions. The use of tailored instructions can link the specific search query content to the requirements and preferences of the user. Instructions typically contain detailed information or document-level characteristics that align with user needs, aiming to enhance the precision and relevance of search results. As shown in Figure 1, we conducted an extensive analysis of real users and their underlying intentions to identify six factors (dimensions) influencing search behaviors: user background (**Audience**), specific search terms or topics (**Keyword**), preferred format for information presentation (**Format**), required response length (**Length**), language requirement (**Language**), and information origin (**Source**). To enhance the diversity of instructions across these six dimensions, we established multiple conditional branches within each dimension, allowing instructions to dynamically adapt and expand based on different conditions. Moreover, even within the same dimension and conditions, we create varied

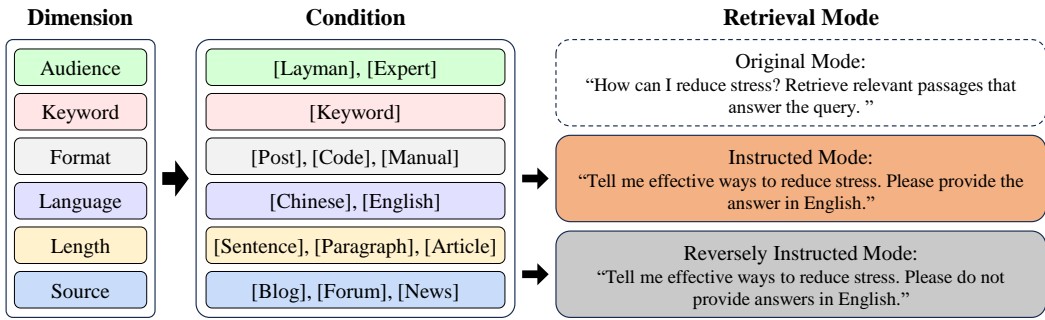

Figure 1: *InfoSearch* consists of six dimensions, each representing a document-level feature with values drawn from predefined conditions. Queries are paired with one dimension and evaluated in three retrieval modes based on the given instructions.

instructions using diverse wording and expressions. This approach not only enriches the dataset but also strengthens the robustness and reliability of the evaluation framework by simulating a broad range of potential user inputs.

Drawing inspiration from (Zhang et al., 2024), we incorporate semantic negation into the dataset by reversing the meaning of instructions across each dimension. This approach allows each query to be associated with multiple instructions, covering various conditions and offering both positive and negative semantic contexts. This ensures that the model is exposed to three distinct retrieval modes:

- **Original Mode**: This mode serves as a baseline that evaluates the model's basic retrieval ability to find pertinent information without any specific constraints.
- **Instructed Mode**: In this mode, the model is required to find documents that are content relevant and satisfy the condition specified in the instruction.
- **Reversely Instructed Mode**: In this mode, the model is required to find documents that are content relevant and do *not* satisfy the condition specified in the instruction, which tests the model's ability to understand negation.

By integrating six dimensions and three distinct retrieval modes, we have developed the comprehensive evaluation dataset *InfoSearch*. This dataset serves as a robust tool for systematically systematically evaluating model's ability to interpret and respond accurately to diverse instructions during retrieval.

## 2.2 CONSTRUCTION PROCESS

The primary objective of developing *InfoSearch* is to bridge queries with diverse instructions, ensuring precise alignment between instructions and their corresponding target documents. We achieve this by collecting Question-Answer (Q-A) pairs for each dimension and expanding the target document pool through web-retrieved content. Figure 2 outlines the 7-step construction process. Data sources, methodological details (e.g., GPT-4 prompts), implementation challenges and dataset statistics are provided in Appendix A.

Step 1: **Condition Determination:** Queries are diversified via multiple conditions, enabling a single query to yield distinct relevant documents depending on contextual requirements.

Step 2: **Data Collection:** To ensure query naturalness and minimize generation costs, we consciously integrate conditions when filtering Q-A pairs from existing datasets or web pages.

Step 3: **Instruction Generation:** Requires producing precise, concise, and natural instructions that reflect natural conversational patterns, aligning with users' tendency to express intents.

Step 4: **Document Rewriting:** When queries or documents inadequately address instruction requirements, GPT-4 refines existing content to produce contextually appropriate documents.

Step 5: **Instruction Reversal:** To verify whether ranking improvements stem from instruction comprehension, instruction semantics are systematically reversed, testing model robustness against persistent high-ranking results.

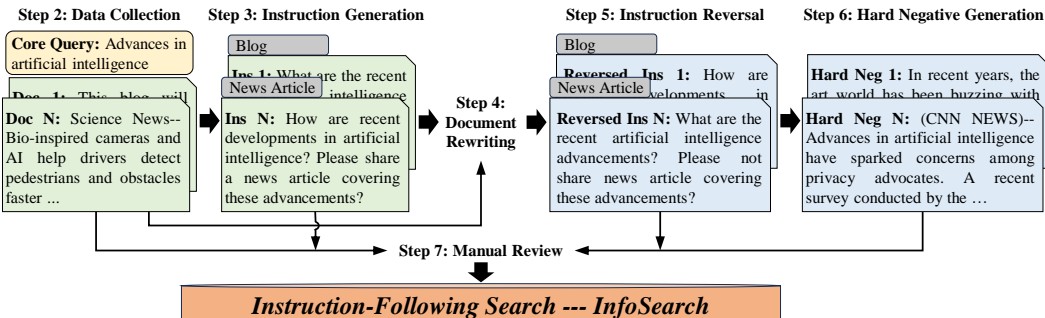

Figure 2: Overview of the dataset construction process for ***InfoSearch***.

Step 6: **Hard Negative Generations:** Adversarial examples are added to resist the model's tendency to depend on superficial document features rather than query-document relationships.

Step 7: **Manual Review:** Anomalous outputs were excluded, prioritizing documents that consistently underperformed or had low relevance scores, followed by expert verification.

By applying the data construction process described above, the ***InfoSearch*** benchmark comprises 600 core queries, 1,598 instructed queries, 1,598 reversely instructed queries, and 6,392 documents.

## 2.3 EVALUATION METRICS

In real-world search systems, user experience hinges on the relevance of top-K results, which directly reflects model efficacy. Thus, instruction-following models must be evaluated based on both original query rankings and their responsiveness to instructions. While metrics like Robustness@k (Oh et al., 2024) and $p$-MRR (Weller et al., 2024a) assess instruction compliance, they exhibit five critical limitations: ① Robustness vulnerability to single anomalies. ② Neglects instruction-response variations. ③ Ignores top-K ranking importance. ④ Insensitive to high-rank changes. ⑤ Inadequate handling of edge cases. A detailed analysis of these limitations is provided in Appendix B.

We define two metrics to quantify the model's responsiveness to instructions: Strict Instruction Compliance Ratio (**SICR**) and Weighted Instruction Sensitivity Evaluation (**WISE**) metric. Assuming that in the original mode, the core query $q$ has $n$ positive documents, denoted as $\mathbf{P} = \{P_1, P_2, \ldots, P_n\}$. When it comes to the instructed mode where the core query is designates a single gold document $\mathbf{P}^i$ out of $\mathbf{P}$, demoting others to negatives. When In the reversely instructed mode, $\mathbf{P}^i$ becomes negative. Let $R_{ori}$, $R_{ins}$ and $R_{rev}$ denote $\mathbf{P}^i$'s rankings and $S_{ori}$, $S_{ins}$ and $S_{rev}$ its relevance scores across original, instructed, and reversed modes.

**Strict Instruction Compliance Ratio**    The **SICR** metric introduces a strict criterion for evaluating sensitivity to instructions. Ideally, for a retrieval result that strictly adheres to the instruction, the gold document's ranking and relevance score in the instructed mode should be higher than in the original mode, denoted as $(R_{ins} < R_{ori} \ \& \ S_{ins} > S_{ori})$. Simultaneously, in the reversely instruction mode, its ranking and relevance score should be lower than those in the original mode, denoted as $(R_{ori} < R_{rev} \ \& \ S_{ori} > S_{rev})$. A query that strictly satisfies these criteria is assigned a score of 1. Implementing rigorous scoring criteria ensures that all changes of relevant documents are taken into account, thereby effectively addressing the issue of low-score sensitivity (defect ①) and and incomplete evaluation (defect ②). The formula for this criterion is as follows:

$$I(q) = \begin{cases} 1, & (R_{ins} < R_{ori}) \text{ and } (S_{ins} > S_{ori}) \text{ and } (R_{ori} < R_{rev}) \text{ and } (S_{ori} > S_{rev}), \\ 0, & \text{otherwise,} \end{cases} \tag{1}$$

The SICR score is calculated as the ratio of the number of queries meeting the instruction-following criteria to the total number of queries in the test set, represented by the following formula:

$$SICR = \frac{\sum_{j=1}^{J} I(q_j)}{|Q|}, \tag{2}$$

Where $|Q|$ represents the total number of queries in the test set. This formula calculate the percentage of retrievals that strictly adhere to the specified instructions relative to the total results.

**Weighted Instruction Sensitivity Evaluation**    The SICR metric reflect the proportion of model results that comply with instructions but lacks a detailed quantification of the degree of compliance. On this basis, the **WISE** metric relaxes the evaluation criteria by focusing only on the ranking changes, regards the results that meet $(R_{ins} \leq R_{ori} < R_{rev})$ [2] as following the instructions, and provides more levels of rewards or penalties for the results. It can be calculated using the following formula:

$$F(q) = \begin{cases} f_{reward}(R_{ori}, R_{ins}), & \text{if } R_{ins} \leq R_{ori} < R_{rev}, \\ f_{penalty}(R_{ori}, R_{ins}), & \text{otherwise,} \end{cases} \tag{3}$$

When defining the reward component, the model is expected to comprehend and execute the instructions, effectively optimizing the rankings of the top K results accordingly. This implies that significant ranking changes within the top K results should be given greater weight to address defects ③ and ④, as these changes are more likely to be noticed and utilized by users. It is essential to consider both the absolute ranking $R_{ins}$ and the relative ranking $(R_{ori} - R_{ins})$. To achieve this, we introduced the $\frac{1}{\sqrt{R_{ins}}}$ term, generously rewarding smaller $R_{ins}$ values. Simultaneously, through the $(1 - \frac{R_{ori}-R_{ins}}{K})$ term, we grant higher rewards to results that demonstrate substantial ranking improvements. Additionally, a uniform value of 0.01 is assigned to results beyond the Top K. More extreme cases are considered (defects ⑤): if a core query contains $N$ positive documents in the original mode and meets the conditions $R_{ori} \leq N$ and $R_{ins} = 1$, it will be granted a reward of 1. This is based on the premise that, for an ideal retriever, the $N$ positive documents would likely rank at the top in the original mode. Accordingly, results that rank higher and exhibit more significant changes should be assigned greater weight. The reward formula is defined as:

$$f_{reward} = \begin{cases} 1, & \text{if } R_{ori} \leq N \text{ and } R_{ins} = 1, \\ (1 - \frac{\sqrt{R_{ori}-R_{ins}}}{K}) \cdot \frac{1}{\sqrt{R_{ins}}}, & \text{if } R_{ori} \leq K, \\ 0.01, & \text{otherwise,} \end{cases} \tag{4}$$

where $K = 20$ signifies that our primary focus is on the top 20 retrieval results. Some of the reward values are visualized in the Figure 5.

For the penalty component, we reference the design of $p$-MRR, emphasizing the magnitude of the ranking drop and apply stricter demerit points for gold documents that experience a larger decline in ranking. However, unlike $p$-MRR, our $R_{ins}$, $R_{ori}$, and $R_{rev}$ yield six possible permutations. To account for the remaining five cases aside from $(R_{ins} \leq R_{ori} < R_{rev})$, we formulated the following penalty formula: [3]

$$f_{penalty} = \begin{cases} -1, & \text{if } R_{rev} < R_{ori} < R_{ins}, \\ \frac{R_{ori}-R_{ins}}{R_{ins}}, & \text{if } R_{ori} \leq R_{ins}, \\ \frac{R_{rev}-R_{ori}}{R_{ori}}, & \text{if } R_{rev} \leq R_{ori}, \end{cases} \tag{5}$$

In summary, for a test set with $J$ queries, the overall evaluation formula can be expressed as:

$$WISE = \frac{\sum_{j=1}^{J} F(q_j)}{J}, \tag{6}$$

## 3    EXPERIMENTS

This section first introduces the experimental settings in §3.1, followed by a description of the overall retrieval results of different types of search models in §3.2. Lastly, it discusses models' performance across individual dimensions in §3.3.

### 3.1    EXPERIMENTAL SETUP

The goal of the benchmark is to determine how effectively retrieval models adjust their retrieval behavior in response to instructions. To thoroughly assess how state-of-the-art retrieval models follow instructions, we evaluate 15 models across four categories of models:

We selected **15 models** representing the four model architectures:

---

[2]$R_{ori}$ may be equal to 1.

[3]$R_{ori} \leq R_{ins}$ covers two cases: $(R_{ori} \leq R_{ins} \leq R_{rev})$ and $(R_{ori} \leq R_{rev} \leq R_{ins})$. Similarly, $R_{rev} \leq R_{ori}$ covers two cases: $(R_{rev} \leq R_{ori} \leq R_{ins})$ and $(R_{rev} \leq R_{ins} \leq R_{rev})$.

Table 1: Performance comparison of different retrieval models averaged over six dimensions. The last three columns display the ranking of the gold document in the original query ($R_{ori}$), and the relative rank change after applying the instructed and reversed instructions.

| Model | nDCG@10 | | | Robustness@10 | | | $p$-MRR ↑ | WISE ↑ | SICR ↑ | Gold Document Rank | | |
|---|---|---|---|---|---|---|---|---|---|---|---|---|
| | Ori | Ins | Rev | Ori | Ins | Rev | | | | $R_{ori}$ | $R_{ins}$ ↓ | $R_{rev}$ ↑ |
| BM25 | 47.5 | 39.1 | 38.5 | 47.5 | 17.7 | 20.9 | 7.0 | -12.0 | 0.0 | 18.4 | 18.0 | 18.3 |
| *Dense Retrieval* | | | | | | | | | | | | |
| Bge-Large-v1.5 | 53.2 | 34.9 | 34.9 | 53.2 | 15.8 | 21.0 | 21.3 | -29.5 | 1.0 | 20.4 | 25.0 | 24.9 |
| E5-Large-v2 | 60.4 | 52.0 | 49.9 | 60.4 | 26.6 | 30.2 | 5.6 | -23.3 | 0.8 | 14.7 | 12.8 | 13.9 |
| Instructor-XL | 62.6 | 38.4 | 39.3 | 62.6 | 17.5 | 23.4 | 30.4 | -29.8 | 2.7 | 30.5 | 30.5 | 36.3 |
| Mistral-ins-v0.2 | 19.4 | 25.5 | 29.2 | 19.4 | 8.5 | 12.7 | -32.4 | -49.2 | 0.0 | 236.0 | 153.0 | 153.1 |
| GTE-Qwen2 | 43.6 | 43.1 | 48.5 | 43.6 | 18.7 | 26.5 | -21.7 | -39.0 | 0.1 | 104.3 | 75.3 | 71.3 |
| E5-Mistral-ins | 78.3 | 64.3 | 66.0 | 78.3 | 41.8 | 46.4 | 4.0 | -16.3 | 2.8 | 6.6 | 5.4 | 5.6 |
| GritLM | 70.8 | 66.2 | 66.3 | 70.8 | 44.2 | 48.3 | -4.3 | -11.1 | 6.9 | 14.4 | 5.8 | 8.9 |
| SFR-Embedding-2-R | 70.7 | 62.2 | 60.1 | 70.7 | 40.7 | 43.2 | 4.8 | -18.1 | 2.1 | 7.4 | 5.7 | 5.6 |
| NV-Embed-v2 | 69.5 | 54.5 | 52.2 | 69.5 | 33.3 | 36.0 | 17.7 | -13.5 | 2.8 | 8.1 | 8.7 | 9.3 |
| *Point-wise Reranking* | | | | | | | | | | | | |
| Mistral-ins-v0.2 | 62.0 | 58.4 | 59.0 | 62.0 | 38.0 | 44.7 | -2.3 | 4.1 | 8.1 | 6.5 | 4.7 | 8.8 |
| Llama-3.1 | 74.8 | 66.8 | 65.4 | 74.8 | 46.1 | 49.2 | 11.5 | 14.4 | 19.3 | 5.4 | 3.7 | 8.2 |
| FollowIR | 72.4 | 66.3 | 65.5 | 72.4 | 46.2 | 50.0 | 4.1 | 13.4 | 12.5 | 5.5 | 3.8 | 7.6 |
| *List-wise Reranking (Fine-tuned)* | | | | | | | | | | | | |
| Zephyr-beta | 70.8 | 55.9 | 58.0 | 70.8 | 32.0 | 36.4 | 1.7 | -3.2 | 8.7 | 6.4 | 6.1 | 7.0 |
| RankVicuna-v1 | 65.4 | 55.2 | 55.2 | 65.4 | 31.2 | 35.7 | 2.0 | -6.5 | 5.6 | 7.3 | 6.3 | 7.0 |
| RankZephyr-v1 | 75.0 | 63.5 | 64.7 | 75.0 | 41.8 | 47.5 | 0.7 | 14.5 | 10.5 | 4.5 | 4.4 | 5.4 |
| *List-wise Reranking (Instructional Zero-shot)* | | | | | | | | | | | | |
| Mistral-ins-v0.2 | 74.5 | 64.4 | 61.6 | 74.5 | 40.5 | 42.2 | 7.2 | 8.1 | 22.0 | 5.7 | 4.8 | 7.2 |
| GPT-4o | 83.8 | 74.2 | 74.2 | 83.8 | 53.0 | 58.0 | 15.0 | 33.5 | 32.1 | 2.6 | 1.7 | 4.3 |

- **Sparse retrieval**: 1 model, BM25 (Robertson et al., 2009).

- **Dense retrieval**: 8 models, including BGE-large-v1.5 (Xiao et al., 2023), E5-large-v2 (Wang et al., 2022), Instructor-XL (Su et al., 2023), E5-Mistral (Wang et al., 2023), GritLM (Muennighoff et al., 2024), NV-Embed-v2 (Lee et al., 2024a), GTE-Qwen2 (Li et al., 2023), and SFR-Embedding-v2 (Meng et al., 2024).

- **Fine-tuned ranking models**: 3 models, including FollowIR (Weller et al., 2024a), RankVicuna (Pradeep et al., 2023a), RankZephyr (Pradeep et al., 2023b), where FollowIR is a point-wise model and the other two are list-wise.

- **Instruction-tuned generation models used for reranking**: 3 models, including Mistral-7B-Instruct-v0.2 (Jiang et al., 2023), Zephyr (Tunstall et al., 2023), and GPT-4o (Achiam et al., 2023).

For dense retrieval models, we compute the dot product between query and document vectors to determine retrieval rankings. For reranking models, the top 100 results from E5-mistral (Wang et al., 2023) are re-ranked based on the models' interpretation of the instruction. For general large language models, we use two settings: In the point-wise setting, both the query and document are inputs, with the output probabilities of *True* or *False* used as similarity scores. In the list-wise setting, following (Pradeep et al., 2023b), a list of documents is provided as a prompt (see Appendix C), and the model returns the ranked document IDs in a list.

Based on Mistral-7B-Instruct-v0.2 (Jiang et al., 2023), we conduct a specialized experiment to evaluate its zero-shot performance in three retrieval settings: dense retrieval, point-wise reranking, and list-wise reranking. As a highly capable instruction-tuned model, Mistral is expected to demonstrate instruction-following abilities in retrieval tasks without fine-tuning. This experiment explores Mistral's potential as a zero-shot retrieval model, assessing whether it can naturally generate strong embeddings or act as an effective reranker to identify instruction-relevant documents from the list of candidates. For the mode of dense retrieval, we use mean pooling to obtain the sentence level representation.

We include GPT-4o as a strong baseline due to its demonstrated instruction-following capabilities and to set a high performance benchmark for all models.

## 3.2 RESULT OVERVIEW

Table 1 provides a detailed comparison of different retrieval models across six dimensions using nDCG@10, Robustness@10, $p$-MRR, WISE, and SICR. The table also includes the average rankings of the golden documents in the instruction mode across the three retrieval models. Notably, almost all models achieved relatively high nDCG, indicating that relying solely on nDCG is insufficient to capture the impact of instructions on ranking changes. Although Robustness can be used for model comparison, it is unable to assess the extent of performance changes before and after instructions because the relevant documents corresponding to the three retrieval modes differ. $p$-MRR can partially reflect the model's responsiveness to different instructions; however, due to the limitations of this metric, the results are not expressed with sufficient accuracy. For instance, according to $p$-MRR evaluations, the instruction-following performance of bge-large-v.5 and Instructor models is significantly better than that of GPT-4o. Meanwhile, the WISE and SICR scores closely align with the ranking changes of the Gold Document and can clearly distinguish the instruction-following capabilities of the models as well as the performance differences between them. The results reveal distinct patterns in instruction-following performance across different model categories, which can be summarized as follows: list-wise reranking models > point-wise reranking models > dense retrieval models > sparse retrieval models. Larger model architectures typically outperform smaller models in both WISE and SICR.

The WISE and SICR scores of the sparse retrieval model BM25 indicate that models relying solely on lexical matching, without sensitivity to instruction-based retrieval or context-aware instructions, struggle to interpret and act on complex instructions. BM25's inability to adapt underscores the limitations of traditional sparse retrieval for instruction-following tasks.

In contrast, dense retrieval models show greater sensitivity to instructions, though their performance varies. For instance, BGE-Large-v1.5 and Instructor-XL demonstrate significant performance degradation under instructions, as reflected in their negative WISE scores. However, models like GritLM, E5-Mistral-ins and NV-Embed-v2 demonstrate greater adaptability. Notably, GritLM achieves the highest WISE and SICR score among the dense models, indicating that, benefiting from joint training on both encoding and generative tasks, GritLM is better equipped to handle complex instructions. In contrast, models primarily trained on task-specific instructions, such as BGE-Large-v1.5 and Instructor-XL, encounter difficulties when addressing a broader range of instructions.

Point-wise reranking models generally outperform dense retrieval models. Among them, Llama-3.1 achieves the highest WISE and SICR scores. Although it has not been specifically fine-tuned for retrieval tasks, Llama-3.1 benefits from its extensive understanding of language, granting it a certain degree of instruction-following capabilities. FollowIR also demonstrates competitiveness; by fine-tuning with content-aware instructions, FollowIR achieves comparable scores to Llama-3.1 with fewer model parameters.

Among list-wise reranking models, GPT-4o performs the best, achieving the highest scores in WISE and SICR across all models, demonstrating its exceptional capability in handling and adhering to complex instructions. Additionally, RankZephyr shows decent performance but remains closer to pointwise re-ranking models in terms of instruction following, possibly due to limitations in its training data. Although Mistral-ins-v0.2 has the second-highest SICR score after GPT-4o, its WISE score is not as remarkable, indicating that while the model can comprehend instructions, it struggles to effectively elevate the rankings of the corresponding documents.

## 3.3 PERFORMANCE ANALYSIS ACROSS DIMENSIONS

The radar plots in Figure 3 offer a visual summary of how different models perform in these dimensions, highlighting their strengths and weaknesses in instruction following. Across all models, both retrieval and reranking models show significant room for improvement. Particularly, certain dimensions – such as format and audience – consistently present challenges. Performance on these remains suboptimal, indicating that models struggle with instructions requiring specific formatting or audience adaptation. The difficulty likely arises from insufficient exposure to structured data formats such as [StackOverflow Post], [Code Snippet], or [Offical Manual], and a lack of nuanced understanding of diverse audience contexts during training.

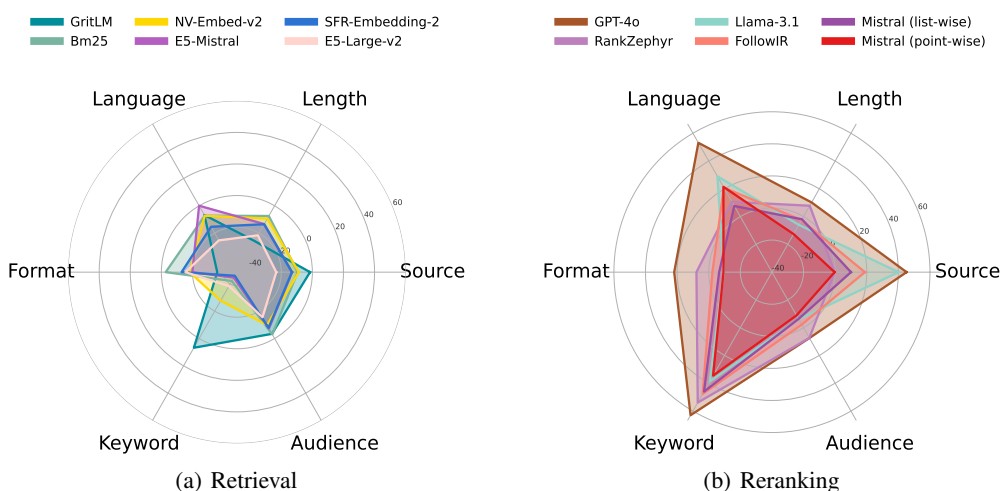

Figure 3: Radar plots comparing the WISE scores of various models across different dimensions, highlighting the strengths and weaknesses of each model in handling different types of instructions. Among retrieval models, GritLM demonstrates the strongest instruction-following capability, while GPT-4 consistently performs the best across all dimensions in the reranking category.

Retrieval models show notable variability in performance across dimensions. GritLM stands out, leading in overall instruction-following ability. Retrieval models generally perform well on the language and keyword dimensions, but they struggle significantly on the format and audience dimensions. This indicates that retrieval models handle text-based instructions effectively but struggle with structural and contextual cues.

Compared to retrieval models, reranking models generally perform better across all dimensions. This improvement is particularly evident in the keyword dimension, largely because reranking models, during inference, directly verify keyword presence within the context. GPT-4 stands out in the language, source, and keyword dimensions, consistently outperforming other models. However, even top-performing models like GPT-4o face challenges with audience-related instructions. Despite the overall performance gap in the Source and Audience dimensions, RankZephyr performs comparably to GPT-4o in the length, audience, and keyword dimensions, demonstrating the effectiveness of fine-tuning for reranking tasks.

## 4 ANALYSIS

$p$-**MRR vs. WISE**. While both metrics aim to measure models' instruction-following abilities by considering rank changes, $p$-MRR does not consistently reflect real performance. Many models in this study received $p$-MRR scores that were inconsistent with the ranking trends of the gold documents; for instance, GPT-4o scored 15.0, which was lower than both Instructor-XL and NV-Embed-v2. Eventually, most models scored even lower than BM25. This discrepancy arises because $p$-MRR evaluates only relative ranking changes ($R_{ins} - R_{ori}$), disregarding absolute ranking shifts ($R_{ori}$). In contrast, the proposed WISE metric strictly enforces instruction following standards by accounting for both absolute and relative ranking changes. GPT-4o achieved the highest WISE score, as it was able to further elevate the rankings of top golde documents when instructions were added and to lower the rankings under semantically inverse instructions ($R_{ori} = 3.51, R_{ins} < R_{ori}, R_{ori} < R_{rev}$).This makes WISE a more reliable metric for evaluating instruction-following capabilities.

**Dense retrieval model vs Reranking model.** Reranking models(represented by red and gray row in Table 2) significantly outperform most dense retrieval models(represented by green row in Table 2) in instruction-following tasks due to their ability to evaluate documents in relation to one another, optimizing the final ranking based on contextual relevance and nuanced understanding. By considering the entire list of retrieved documents, rerankers can effectively adjust the order based on the specific needs of the query, capturing subtle distinctions that dense retrieval models may overlook. This leads to more accurate rankings that align with user intent, especially in

Table 2: Performance comparison of different retrieval models across six dimensions using the WISE and SICR metrics. The dimensions are: D1 (Audience), D2 (Keyword), D3 (Format), D4 (Language), D5 (Length), and D6 (Source). Higher scores indicate stronger instruction-following capabilities.

| Model | WISe | | | | | | | SICR | | | | | | |
|---|---|---|---|---|---|---|---|---|---|---|---|---|---|---|
| | D1 | D2 | D3 | D4 | D5 | D6 | Avg. | D1 | D2 | D3 | D4 | D5 | D6 | Avg. |
| BM25 | -3.0 | -42.1 | -2.8 | -7.2 | -7.5 | 1.97 | -12.0 | 0.0 | 0.0 | 0.0 | 0.0 | 0.0 | 0.0 | 0.0 |
| *Dense Retrieval* | | | | | | | | | | | | | | |
| Bge-Large-v1.5 | -16.8 | -38.2 | -42.1 | -20.7 | -28.6 | -30.7 | -29.5 | 0.5 | 0.0 | 0.3 | 2.0 | 1.0 | 2.3 | 1.0 |
| E5-Large-v2 | -15.6 | -38.3 | -15.5 | -25.3 | -21.6 | -23.2 | -23.3 | 1.4 | 0.7 | 0.0 | 0.5 | 0.7 | 1.3 | 0.8 |
| Instructor-XL | -27.7 | -34.7 | -30.5 | -20.5 | -35.5 | -29.8 | -29.8 | 5.7 | 2.1 | 0.3 | 4.0 | 0.3 | 4.0 | 2.7 |
| Mistral-ins-v0.2 | -35.8 | -67.8 | -29.7 | -31.9 | -66.6 | -63.3 | -49.2 | 0.0 | 0.0 | 0.0 | 0.0 | 0.0 | 0.0 | 0.0 |
| GTE-Qwen2 | -34.0 | -36.5 | -44.3 | -18.0 | -56.4 | -44.6 | -39.0 | 0.0 | 0.0 | 0.0 | 0.0 | 0.3 | 0.0 | 0.1 |
| E5-Mistral-ins | -7.3 | -44.5 | -19.9 | 0.1 | -13.4 | -13.0 | -16.3 | 2.9 | 0.0 | 0.0 | 10.5 | 0.0 | 3.3 | 2.8 |
| GritLM | -3.4 | 6.8 | -36.0 | -6.7 | -25.8 | -1.5 | -11.1 | 11.4 | 11.8 | 1.3 | 4.5 | 0.3 | 11.7 | 6.9 |
| SFR-Embedding-2-R | -7.8 | -45.9 | -13.0 | -15.4 | -13.5 | -13.2 | -18.1 | 2.9 | 1.0 | 2.0 | 1.0 | 1.0 | 4.7 | 2.1 |
| NV-Embed-v2 | -9.8 | -27.7 | -18.1 | -7.3 | -9.3 | -8.7 | -13.5 | 2.4 | 0.7 | 1.0 | 3.5 | 0.3 | 9.0 | 2.8 |
| *Point-wise Reranking* | | | | | | | | | | | | | | |
| Mistral-ins-v0.2 | -8.9 | 34.5 | -9.3 | 21.4 | -12.6 | -0.3 | 4.1 | 1.4 | 28.6 | 2.7 | 6.5 | 4.7 | 4.7 | 8.1 |
| Llama-3.1 | -6.2 | 38.7 | -9.5 | 29.0 | -5.9 | 40.2 | 14.4 | 6.2 | 38.3 | 10.0 | 22.0 | 2.7 | 36.7 | 19.3 |
| FollowIR | -2.3 | 47.7 | -2.3 | 20.9 | -2.6 | 18.8 | 13.4 | 3.3 | 27.2 | 7.0 | 19.5 | 1.7 | 16.3 | 12.5 |
| *List-wise Reranking* | | | | | | | | | | | | | | |
| Mistral-ins-v0.2 | -6.3 | 46.0 | -6.6 | 7.6 | -1.9 | 10.0 | 8.1 | 10.5 | 59.2 | 9.7 | 23.0 | 8.0 | 21.7 | 22.0 |
| Zephyr-beta | -2.7 | 14.1 | -13.9 | -6.9 | -5.7 | -3.9 | -3.2 | 1.0 | 27.5 | 8.0 | 10.5 | 2.0 | 3.0 | 8.7 |
| RankVicuna-v1 | -2.5 | -8.5 | -9.8 | -11.8 | -4.3 | -2.2 | -6.5 | 5.2 | 10.5 | 3.3 | 4.5 | 2.3 | 8.0 | 5.6 |
| RankZephyr-v1 | 7.4 | 53.9 | 7.8 | 10.6 | 7.8 | -0.3 | 14.5 | 4.3 | 42.5 | 1.0 | 5.5 | 4.3 | 5.3 | 10.5 |
| GPT-4o | 7.4 | 63.0 | 21.9 | 53.1 | 10.2 | 45.2 | **33.5** | 15.2 | 60.6 | 11.3 | 55.5 | 10.3 | 39.7 | **32.1** |

complex scenarios where the relationship between documents and the query is crucial for effective instruction-following. Consequently, while dense retrieval models excel in efficiently retrieving relevant documents, reranking models provide the precision necessary to enhance the overall ranking quality, resulting in superior performance in tasks requiring sophisticated language comprehension.

**Point-wise reranking vs. List-wise reranking.** Point-wise reranking evaluates each document independently, predicting relevance without considering other documents. In contrast, list-wise reranking considers the entire set, optimizing the overall ranking order by leveraging relative relationships between documents. As shown in Table 2, list-wise ranking (gray row) generally outperforms point-wise ranking (red row) for instruction-following tasks, as it better captures the broader query context and the relative importance of documents. This makes list-wise reranking more effective in organizing documents to align with complex instructions, improving relevance and coherence across different queries.

**Zephyr vs. RankZephyr.** RankZephyr outperforms Zephyr in both WISE and SICR because of its more sophisticated training process, better robustness to initial document order, multiple reranking passes that help correct ranking errors. Compared to Zephyr, RankZephyr learns from RankGPT, which allows it to adopt more sophisticated ranking strategies. Besides, RankZephyr benefits from multiple passes, allowing it to adjust the ranking more effectively compared to a single-pass strategy like Zephyr's, which might not optimize the ranking as thoroughly. These factors combine to ensure that RankZephyr minimizes penalties for ranking important documents too low, leading to significantly better WISE and SICR scores.

**Mistral (retrieval) vs. (point-wise) vs (list-wise).** Mistral-ins-v0.2 shows poor retrieval performance, highlighting its limitations in handling complex, instruction-driven ranking scenarios. Without specific training for retrieval, it fails to rank documents effectively on nuanced instructions. On the other hand, Mistral-ins-v0.2 in point-wise ranking demonstrates improved performance in both WISE and SICR, as it scores individual documents independently, allowing it to better adhere to instructions, though it lacks the depth to consider relationships between documents. However, Mistral-ins-v0.2 in list-wise ranking truly excels, as it optimizes the entire list and takes document interactions into account, enabling it to handle more sophisticated instruction-following tasks. This results in significantly better WISE and SICR scores, making list-wise Mistral-ins-v0.2 the most effective approach for instruction-following tasks where ranking coherence is critical.

## 5 RELATED WORK

**Dense Retrieval**   The development of dense retrieval models has significantly enhanced the semantic understanding and efficiency of retrieval systems. Existing dense models can be categorized into two types based on their architecture: Bidirectional Embedding Models and Decoder-only Embedding Models. Bidirectional Embedding Models are typically base on BERT (Devlin, 2018) or T5 (Raffel et al., 2020) encoders, performing general embedding tasks. Early models that base on BERT or T5 for efficient text embeddings include Sentence BERT (Reimers, 2019), SimCSE (Gao et al., 2021) and Sentence T5 (Ni et al., 2021). To better accommodate the requirements of text embeddings, researchers have pre-trained these encoders using contrastive learning (Izacard et al., 2021; Wang et al., 2022).Furthermore, these models are fine-tuned using various supervised datasets to enhance their performance in retrieval tasks or other downstream applications (Lee et al., 2024b; Li & Li, 2023). Compared to bidirectional embedding models, decoder-only embedding models initially perform relatively poorly in general embedding tasks, primarily due to their limited capacity to comprehensively capture and utilize contextual information (Brown, 2020). However, many researchers have sought to optimize these models' performance by introducing contrastive learning methods to address their deficiencies in embedding tasks (Neelakantan et al., 2022). Currently, researchers have explored not only the use of synthetic data (Wang et al., 2023) but also a hybrid strategy combining real and synthetic data (Meng et al., 2024; BehnamGhader et al., 2024), achieving significant success in text embedding tasks. Collectively, these advancements in contrastive pre-training, model scaling, and leveraging weak supervision and synthetic data significantly propel the field of retrieval.

**Instruction-Following for Retrieval**   The notion of relevance often varies among users (Mizzaro, 1998). Consequently, queries alone may not fully address all users' information needs (Ruthven & Lalmas, 2003), while instructions can expand these intentions beyond the scope of the queries. Recent information retrieval research has recognized this and tried to train retrieval models by combining instructions with queries to enhance their instruction-following capabilities. In general, existing instruction-following models can be divided into two categories based on instruction design methods: task-aware and content-aware instruction retrieval models. TART first proposed a general retrieval system with task-level instruction, setting specific instructions for different retrieval tasks to query corresponding results (Asai et al., 2023). Subsequently, Instructor expanded the scope of instructions so that text embeddings can not only retrieve but also classify and diagnose duplicate problems (Su et al., 2023). However, these task-aware instructions are too general and lack the specificity of user instructions in real scenarios. On this basis, other researchers have developed content-level instructions. InstructIR set instructions to adapt to query-text pairs based on user background (such as work, hobbies) (Oh et al., 2024). ExcluIR set exclusionary instructions based on the content differences between query results, accounting for users' exclusionary needs in queries (Zhang et al., 2024). FollowIR set instructions to distinguish query results by combining exclusion and inclusion (Weller et al., 2024a). PIR focuses on the ability of retrievers to recognize and respond to different perspectives in queries (Zhao et al., 2024). However, real user intentions involve both internal (content, audience, language) and external (format, length) answer attributes. MAIR proposed a large-scale instruction retrieval benchmark covering 126 different information retrieval tasks, but lacks an explicit evaluation of the model's instruction following ability (Sun et al., 2024).

## 6 CONCLUSION

Despite leveraging LMs as the backbone for training retrieval models, most existing IR models cannot truly understand the instructions in query. Further, traditional score indicators(e.g., nDCG) cannot reflect whether the model has the ability to follow instructions and most existing dataset with instructions are designed with a only single dimension, so we propose ***InfoSearch*** and two novel metrics(WISE, SICR). The choice of dimensions in our dataset takes into account the instructions that users may give in actual scenarios. Additionally, our metrics consider the combined performance of the model in three modes(Original mode, Instructed mode, Reversely instructed mode), with increasing difficulty across modes as each introduces more complex challenges. We hope this work helps the future instruction-following retrieval task.

ACKNOWLEDGEMENT

We thank EIT and IDT High Performance Computing Center for providing computational resources for this project. This work is supported by 2035 Key Research and Development Program of Ningbo City under Grant No.2024Z127.

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

# A  MORE DETAILS OF INFOSEARCH

## A.1  CONDITION DETERMINATION

The query for each dimension can be diversified and expanded through various conditions, allowing the same query to correspond to different relevant documents under different conditions. Except for the Keyword dimension, the other five dimensions have fixed conditions. The condition of the Keyword dimension is the keyword in the document that is relevant to the query. Therefore, the condition of the Keyword dimension needs to be determined after filtering out the Query-Document (Q-D) pairs. To achieve this, both the query and its corresponding document were input into GPT-4, generating a unique condition for each Q-D pair. However, GPT-4 occasionally selected irrelevant words, such as "and" or "what", that did not align with the user scenario when generating keyword conditions. To address this, we meticulously crafted prompt templates (Table 3) for condition extraction, ensuring that the conditions were both unique and representative of each document, accurately reflecting the document's core theme.

Table 3: A template that generates the specific conditions required for the keyword dimension

| **Prompt for Condition Generation** |
| --- |

### TASK ###
- Your task is to generate a condition that refines a given query in relation to a provided document. The condition should be:
    1. Relevant to the document's core topic – It must align with the central theme or key content of the document.
    2. A meaningful constraint on the query – It should introduce a specific aspect, subtopic, or perspective that naturally extends the original query while still being directly supported by the document.
    3. Not a generic or arbitrary restriction – The condition must be logically derived from the document's content and should not be a trivial or overly broad constraint.

### INPUT ###
- You will receive a query and a document as input:
    – Query: {query}
    – Document: {document}

### FORMATTING ###
- Condition: <the condition your generated >

## A.2  DATA COLLECTION

To ensure the queries realistic and reduce the human cost, we consciously integrate conditions when filtering Q-A pairs from existing datasets or web pages. For instance, in the Format dimension, due to the lack of available multi-format Q&A datasets, we selectively extract Q-D pairs from StackOverflow posts. For posts containing code and detailed official documentation responses, we use their titles as queries and the complete responses as documents under the [StackOverflow] condition. The pure code snippets within the answers and references to official documentation are separately extracted and used as documents under the [Code Snippet] and [Manual] conditions, respectively. Table 4 shows the source of datasets used to collect query-document pairs for each dimension.

## A.3  INSTRUCTION GENERATION

The generation of accurate, concise, and natural instructions is crucial. When searching, users tend to express their intentions using simple, naural language, so the generated instructions must remain

Table 4: Structure and source of the dataset

| Dimension | Source Data | Condition Value |
|---|---|---|
| Audience | BioASQ, scifact (Muennighoff et al., 2022) | [Layman], [Expert] |
| Keyword | MSMARCO (Bajaj et al., 2016) | [keyword] |
| Format | Stackoverflow, various office doc | [Stackoverflow Post], [Code Snippet], [Official Manual] |
| Language | publichealth-qa | [Chinese], [English] |
| Length | medical_qa (Muennighoff et al., 2022), google search | [Sentence], [Paragraph], [Article] |
| Source | CNN-english-news, google search | [Blog], [Forum Post], [News Article] |

brief and clear, closely mirroring conversational style. To achieve this, we employed words such as "smooth", "natural", and "realistic" in the prompts (see Table 5) to guide GPT-4 in crafting instructions that emphasize not only semantic accuracy but also the emulation of authentic user expressions. Furthermore, a two-sentence structure for the instructions, first rephrasing the query and then appending specific constraints. This structure effectively separates the core query from the conditions, enhancing the diversity of generated instructions. For example, "What are the most effective exercises for losing weight? Please find discussions from forum posts only." This two-sentence structure ensures logical clarity and semantic coherence.

Table 5: A prompt template for Generating Instruction

**Prompt for Instruction Generation**

### TASK ###
- You are tasked with generating a natural query with an instruction based on the query and the condition provided by the user. You will be provided with a query and a condition and you need to:
    1. Rephrase the core query as the first sentence, making it sound like a natural human query without changing its meaning.
    2. Create a second sentence that specifies the search restriction.
    3. Ensure each sentence is smooth, concise, reasonable, natural, and realistic, mimicking a real human tone.

### INPUT ###
- Core Query: {core query}
- Condition: {condition}

### FORMATTING ###
- Core query: <the core query I give you >
  Condition: <the condition I give you >
  Query_with_Instruction: <the query with instruction you generated >

## A.4 DOCUMENT REWRITING

When the query and relevant documents fail to meet the instruction requirements, we use GPT-4 to rewrite the existing documents to generate relevant documents. The documents that need to be modified are mainly concentrated in the source, length and audience dimensions, so we set specific prompts for these dimensions respectively (see Table 6). In this process, we experimented with directly generating condition-satisfying documents from the query, but these often exhibited redundant expressions and inconsistent formatting. Therefore, we adjusted the existing documents, ensuring they meet the instruction requirements while preserving naturalness and authenticity in the language.

Table 6: Prompt templates for document rewriting

| Dimension | Prompt Template |
|---|---|
| Source | **### TASK ###**

• For a core query, I need documents from a blog, forum post, or news article. I will provide you with a core query, the corresponding document from a news article. Your task is to rewrite the document as blog and forum post content.

**### CAUTION ###**

1. For the blog you generated, you cannot use the core query as blog title directly. You need to rephrase it, but do not change the semantics of this query. Besides, you need to give various information in the line under the title, such as the author, when it was published, the word "Blog", and the section it belongs to. All the above information must be random.

2. For a forum post, it must be a form of discussion among multiple users. The usernames need to be random rather than use "use1", "use2" etc.

**### INPUT ###**

• Core Query: {core query}
• Document: {document}

**### FORMATTING ###**

• Your output should be in the following format:
• Blog: <the blog you generated >
Forum: <the forum post you generated > |
| Audience | **### TASK ###**

• I will provide you with a core query and its corresponding document. The target audience for this document is experts. Your task is to Rewrite this document to make it easily understandable for laymen.

**### CAUTION ###**

1. Keep the semantics of the document intact.

2. Do not use any technical jargon in the rewritten document for layman.

**### INPUT ###**

• Query: {query}
• Document for expert: {expert}

**### FORMATTING ###**

• Query: <the query I give you >
Layman: <the rewritten document for layman you generated > |
| Length | **### TASK ###**

• I will provide you with a core query and its corresponding paragraph. Your task is to rewrite this paragraph into a single sentence and an article.

**### CAUTION ###**

1. Ensure that both the sentence and article retain the original meaning of the paragraph.

**### INPUT ###**

• Core Query: {core query}
• Paragraph: {paragraph}

**### FORMATTING ###**

• Your output should be in the following format:
• Sentence: <the rewritten single sentence that answers the query >
Article: <the multi-paragraph rewritten document that answers the query> |

## A.5 INSTRUCTION REVERSAL

In real retrieval, we observed that results already ranked highly tend to remain at the top even after instructions are applied. This makes it difficult to determine whether the improvement in ranking is due to the model's understanding of the instructions or simply a result of detailed keyword and semantic matching. To address this, we validate the model's comprehension of the instructions by reversing the semantic meaning of the instructions. For example, "Please answer in Chinese" is reversed to "Please do not answer in Chinese."

Table 7: A prompt Template for Instruction Reversing

| **Prompt for Instruction Reversing** |
| --- |
| **### TASK ###**
• Your expertise lies in interpreting and transforming direct instructions into their opposite or negative forms while maintaining clarity and coherence in the transformed instructions. Your task is to reverse the instruction I give you.

**### CAUTION ###**
• While reversing the instruction, ensure that the new instruction conveys the opposite meaning accurately. Please keep in mind that the transformation should remain clear and easy to understand, avoiding any ambiguity.

**### INPUT ###**
• Instruction: {instruction}

**### FORMATTING ###**
• Reverse Instruction: <the instruction your reverse > |

## A.6 HARD NEGATIVE GENERATIONS

While positive documents for the same query under varying conditions may act as negative examples for one another (instruction negatives), we still need to prevent the model from relying solely on prominent features for simple retrieval, thereby neglecting the subtle relationships between the query and the documents. To address this, we use GPT-4 to generate documents that appear to be related to the query topic on the surface but cannot actually answer the query, serving as hard negative documents (query negatives).

Table 8: A prompt templates for generating hard negative

**Prompt for Hard Negative Generation**

### TASK ###

- You are tasked with generating a hard negative document based on a given query. A hard negative document should appear superficially relevant to the query but contain critical inaccuracies, misleading details, or subtle contradictions. Follow these steps:
    1. Understand the core intent of the query and identify key entities, relationships, or requirements.
    2. Generate a document that incorporates some keywords from the query but does not provide a direct or indirect answer to the query. The document should maintain a plausible structure and stay on a related topic while ensuring that no information within it can be used to infer or construct a correct response to the query.
    3. Ensure the document is coherent, natural, and realistic, mimicking a genuine but incorrect response.

### INPUT ###

- Core Query: {core query}

### FORMATTING ###

- Core query: <the core query I give you >
  Hard negative document: <generated document >

## A.7   MANUAL REVIEW

We filtered out anomalous documents from the outputs of 12 retrieval models, selecting those that failed to rank within the top 50 in six or more models or had a relevance score below 0.5 for the query, followed by manual screening. This process aimed to eliminate mislabeled Q-D pairs selected from other datasets or documents inaccurately retrieved through manual search. For these mismatched documents, we proceed with manual replacement. After multiple rounds of screening to ensure the quality of the InfoSearch dataset, the statistical results are summarized in Table 9.

Table 9: **InfoSearch** dataset statistics. $|Q|$, $|I|$ and $|R|$ represent the word lengths of the original query, instructed query and reversely instructed query respectively.

| Dimension | # $Q$ | Avg $|Q|$ | # $I$ | Avg $|I|$ | # $R$ | Avg $|R|$ | # $D$ |
|---|---|---|---|---|---|---|---|
| Audience | 100 | 9.02 | 210 | 20.46 | 210 | 15.91 | 840 |
| Keyword | 100 | 6.30 | 288 | 17.90 | 288 | 18.92 | 1152 |
| Format | 100 | 9.16 | 300 | 16.65 | 300 | 19.31 | 1200 |
| Language | 100 | 8.75 | 200 | 14.09 | 200 | 15.74 | 800 |
| Length | 100 | 8.52 | 300 | 15.94 | 300 | 16.26 | 1200 |
| Source | 100 | 7.38 | 300 | 18.19 | 300 | 15.58 | 1200 |
| **Total** | 600 | | 1598 | | 1598 | | 6392 |

To make the data more intuitive, Table 10 to Table 15 provide specific examples from each dimension in the ***InfoSearch*** dataset.

Table 10: An example in Audience dimension

| | |
|---|---|
| Core Query | How to Prevent Heart Disease |
| Instructed 1 | Explore effective strategies for preventing heart disease. Please explain in terms that are easy for the general public to understand. |
| Instructed 2 | Investigate the latest preventive measures against heart disease. Make a detailed discussion suitable for a professional audience. |
| Reversed 1 | How to Prevent Heart Disease. I'm looking for a response that is more technical than layman. |
| Reversed 2 | How to Prevent Heart Disease. Please keep your answer simple and clear. |
| Document 1 | To prevent heart disease, consider the following strategies : 
 Adopt a Vegan Diet: Vegan diets, particularly those rich in soy and other plant-based proteins, can reduce the risk of cardiovascular disease. These proteins are high in non-essential amino acids, which promote glucagon activity. Glucagon helps regulate lipid levels and cholesterol synthesis, leading to healthier heart conditions. 
 Increase Glucagon Activity: ... |
| Document 2 | ... Vegan proteins may reduce risk of cancer, obesity, and cardiovascular disease by promoting increased glucagon activity. ... glucagon promotes (and insulin inhibits) cAMP-dependent mechanisms that down-regulate lipogenic enzymes and cholesterol synthesis, while up-regulating hepatic LDL receptors and production of the IGF-I antagonist IGFBP-1. The insulin-sensitizing properties of many vegan diets–high in fiber, low in saturated fat ... |

Table 11: An example in Keyword dimension

| Core Query | What helps for acne? |
| --- | --- |
| Instructed 1 | What treatments are effective for acne? Ensure your answer includes information specifically about "progesterone". |
| Instructed 2 | Can you tell me what helps reduce acne symptoms? Focus on the effects of "mint" in your response. |
| Instructed 3 | What natural remedies are beneficial for managing acne? Please include details about "Chamomile". |
| Reversed 1 | What helps for acne? Can you provide a response that does not involve the term "progesterone"? |
| Reversed 2 | What helps for acne? Can you give me a reply that does not entail the use of the term "mint"? |
| Reversed 3 | What helps for acne? Can you provide a response avoiding the term "Chamomile"? |
| Document 1 | Progesterone helps with acne that occurs in the late 30's and early 40's. Also, if the acne varies with the period, elimination of xenoestrogens (environmental estrogens) and phytoestrogens and taking progesterone cream helps with this type of acne as well. |
| Document 2 | Acne home remedy: Mint. Mint can help remove pore-clogging oil. To help clear acne before it begins, mix 2 tablespoons of finely chopped fresh mint with two tablespoons each of plain yogurt and oatmeal (use a blender to pulverize the oatmeal to powder). Leave the concoction on your face for 10 minutes, then rinse off with water. |
| Document 3 | Acne home remedy: Chamomile. Chamomile helps decrease inflammation from acne. In a blender or coffee grinder, combine the contents of a chamomile tea bag with enough water to form a paste, and apply that to acne. Alternately, steep two chamomile tea bags with 1 cup boiled water for 15 minutes. |

Table 12: An example in Format dimension

| Core Query | How can I access environment variables in Python? |
|---|---|
| Instructed 1 | How can I access environment variables in Python? Limit the search to Stackoverflow posts. |
| Instructed 2 | How can I access environment variables in Python? I need code snippets to solve the problem. |
| Instructed 3 | How can I access environment variables in Python? Only consider official manuals. |
| Reversed 1 | How can I access environment variables in Python? Provide me with an answer that is not a Stackoverflow post.. |
| Reversed 2 | How can I access environment variables in Python? Could you deliver a response that isn't in the form of a code snippet? |
| Reversed 3 | How can I access environment variables in Python? I'm seeking a reply that isn't an official manual. |
| Document 1 | Environment variables are accessed through ['os.environ'] |
| | ```python\nimport os\nprint(os.environ['HOME'])\n``` |
| | To see a list of all environment variables: |
| | ```python\nprint(os.environ)\n``` |
| | If a key is not present, attempting to access it will raise a 'KeyError'. To avoid this: |
| | ```python\n# Returns 'None' if the key doesn't exist\nprint(os.environ.get('KEY_THAT_MIGHT_EXIST'))\n``` |
| Document 2 | ```python\nimport os\nprint(os.environ['HOME'])\n``` |
| Document 3 | os.**environ**
A [mapping] object where keys and values are strings that represent the process environment. For example, 'environ['HOME']' is the pathname of your home directory (on some platforms), and is equivalent to 'getenv("HOME")' in C
This mapping is captured the first time the ['os'] module is imported, typically during Python startup as part of processing 'site.py'. Changes to the environment made after this time are not reflected in ['os.environ'] except for changes made by modifying ['os.environ'] directly.
...
On Windows, the keys are converted to uppercase. This also applies when getting, setting, or deleting an item. For example, 'environ['monty'] = 'python'' maps the key ''MONTY'' to the value ''python''. |

Table 13: An example in Language dimension

| Core Query | What is diabetes? |
|---|---|
| Instructed 1 | Tell me what diabetes is.Please use Chinese. |
| Instructed 2 | Tell me the answer to what is diabetes.Please use English. |
| Reversed 1 | What is diabetes? Please respond in a language other than Chinese. |
| Reversed 2 | What is diabetes? I'd rather have a response in a language other than English. |
| Document 1 | 糖尿病（拉丁语：diabetes mellitus，缩写为DM，简称diabetes）是一种代谢性疾病，它的特征是患者的血糖长期高于标准值。高血糖会造成俗称"三多一少"的症状：多食、多饮、多尿及体重下降。对于第1型糖尿病，其症状会在一个星期至一个月期间出现，而对于第2型糖尿病则较后出现。不论是哪一种糖尿病，如果不进行治疗，可能会引发许多并发症。急性并发症包括糖尿病酮酸血症与高渗透压高血糖非酮酸性昏迷；严重的长期并发症则包括心血管疾病、中风、慢性肾脏病、糖尿病足、以及视网膜病变等；其中糖尿病和心衰竭、慢性肾脏病有着较紧密的共病关系。 |
| Document 2 | Diabetes is a chronic disease that occurs either when the pancreas does not produce enough insulin or when the body cannot effectively use the insulin it produces. |

Table 14: An example in Length dimension

| Core Query | How many calories are in a martini? |
|---|---|
| Instructed 1 | How many calories are in a martini? Please give me a sentence answer. |
| Instructed 2 | What's the calorie count of a martini? I'd like a paragraph explaining it. |
| Instructed 3 | Can you tell me the calories in a martini? Please provide a detailed article. |
| Reversed 1 | How many calories are in a martini.Please provide a detailed response, not just a single sentence. |
| Reversed 2 | How many calories are in a martini.Please avoid giving me a paragraph as your response.? |
| Reversed 3 | How many calories are in a martini.Please don't structure your answer as an article. |
| Document 1 | 2.25 oz (67 mL) Martini (extra dry): 140 calories. |
| Document 2 | The amount of Calories in a martini cocktail can vary based on how you make it. A martini cocktail technically only has two ingredients, vodka and vermouth, so Calorie count depends on your proportions. GREY GOOSE® Vodka contains 66 Calories per 30 ml serving*. Try mixing up our Classic Dry Vodka Martini Cocktail recipe. |
| Document 3 | Vodka Martini Calories
Depending on the size of your cocktail, and the extras you mix in, one serving of a vodka martini is approximately 202 calories. Vodka martini calories can be much higher if the drink has more than the two basic liquors.
To figure out the calories in vodka ... 1 teaspoon of French vermouth has approximately 7.8 calories ... |

Table 15: An example in Source dimension

| Core Query | Effective exercises for weight loss |
| --- | --- |
| Instructed 1 | What's the best way to do exercises for weight loss effectively? Please provide a blog post on this topic. |
| Instructed 2 | How can I perform exercises effectively for weight loss? I'd like a forum post on this subject. |
| Instructed 3 | Tell me how to do effective exercises for weight loss. Give me something from News Articles. |
| Reversed 1 | Effective exercises for weight loss.Please provide a response that is not from a blog. |
| Reversed 2 | Effective exercises for weight loss.I'm looking for an answer that's not based on a forum thread. |
| Reversed 3 | Effective exercises for weight loss.Please avoid using a news article as your source.. |
| Document 1 | What Are the Best Exercises for Weight Loss?
May 6, 2024 Blog
Losing weight can be a challenging journey, but incorporating exercise into your routine can make a significant difference. Not only does exercise help you burn calories, but it also boosts your metabolism, improves your mood and increases your overall health and well-being.
But with so many different types of exercises out there, it can be overwhelming to figure out which ones are the best for weight loss.
How to Exercise for Weight Loss
Walking exercise for weight loss
Walking is a low-impact exercise that is perfect for beginners ... |
| Document 2 | superMario_Milt:
I myself enjoy going on long walks (anywhere from 30 minutes to 2 hours). It's easy on the joints, I can listen to music or stick to my thoughts, and you get fresh air away from being cooped up in a gym. It definitely as helped me trim up some over time.
Individual_Ad_2701:
I do 1-2 hours of lifting a day hate cardio well After lifting I do how much should I walk after I lift like would 20-30 minutes work I'm gaining muscle and I can see that my arms and chest are bigger but my belly is getting bigger also I did try eating less calories but idk.
Proudscobi:
If you are going to choose one for weight loss, go for weight lifting. It will improve your body composition. Even if you don't lose weight you will look better. |
| Document 3 | NBC HEALTH NEWS——Morning workouts may be better for weight loss, study finds. People who got their exercise in between 7 a.m. and 9 a.m. had lower BMIs than those who opted to exercise later in the day. Is morning the best time of day to exercise? Research published Tuesday in the journal Obesity finds that early morning activity — between 7 a.m. and 9 a.m. — could help with weight loss. "My cautious suggestion from this study is that if we choose to exercise in the early morning, before we eat, we can ... |

Moreover, Figure 4 illustrates the distribution of data across six dimensions in the FollowIR, InstructIR, and ***InfoSearch*** datasets. The chart highlights the varying proportions of query-document pairs based on dimensions like Audience, Keyword, Language, Length, Source, and Format.

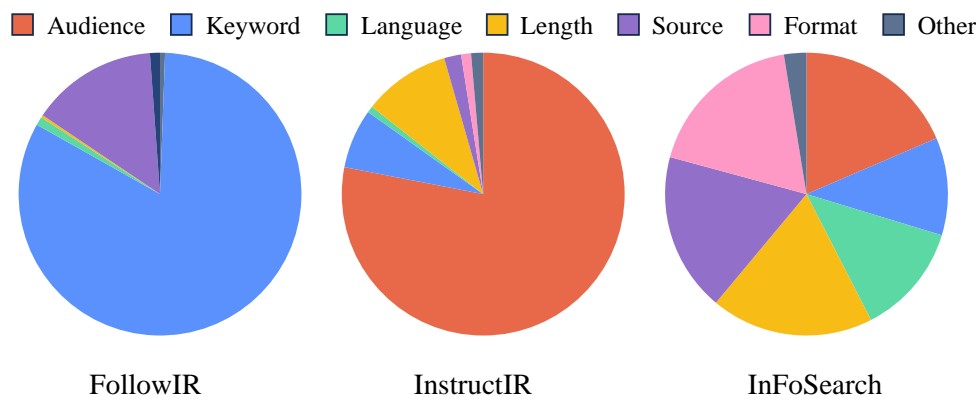

Figure 4: Comparison of the ***InfoSearch*** dataset with FollowIR and InstructIR in terms of data distribution across six dimensions.

## B    EVALUATION METRICS ANALYSIS

To measure instruction-following performance for retrieval models is a challenge. Two metrics were specifically proposed in previous studies for this purpose: Robustness@k (Oh et al., 2024) and $p$-MRR (Weller et al., 2024a). We argue that neither of them effectively reflects true instruction-following performance.

Robustness@k is designed to assess a model's performance on the same query under different instructions. Specifically, it groups instances of the same query, calculates the minimum nDCG@k score within each group, and averages the group scores to generate the overall Robustness@k score. Let $Q = \{q_1, q_2, ..., q_n\}$ be a set of queries. For each query $q_i$, there are $m_i$ distinct instruction variants $\{I_{i1}, I_{i2}, ..., I_{im_i}\}$, calculate the minimum nDCG@k score across all its instruction:

$$min\text{-nDCG}(q_i) = \min_{j \in (1,2,...,m_i)} s_{ij} \tag{7}$$

where $s_{ij}$ represents the nDCG@k score for query $q_i$ under instruction $I_{ij}$. Compute the overall Robustness@k score as the average of these minimum scores across all queries:

$$Robustness@k = \frac{1}{n} \sum_{i=1}^{n} min\text{-nDCG}(q_i) \tag{8}$$

However, the Robustness@k metric oversimplifies the evaluation of a model's ability to follow instructions. ① Even if a model demonstrates strong performance across the majority of queries, a single anomalously low score can reduce the overall robustness score. ② Furthermore, focusing solely on the lowest score disregards variations in the model's responses to different instructions, thus failing to capture the overall performance trend. [4]

As for $p$-MRR, it is based on the MRR metric and quantifies the model's ability to follow instructions by comparing the rankings of relevant documents in the original mode and the instruction mode. The following formula is applied to calculate the score for each relevant document within a query:

$$p\text{-MRR} = \begin{cases} \frac{MRR_{og}}{MRR_{new}} - 1, & \text{if } R_{og} > R_{new} \\ 1 - \frac{MRR_{new}}{MRR_{og}}, & \text{otherwise,} \end{cases} \tag{9}$$

---

[4]For instance, the nDCG@k scores for group A are $\{0.8, 0.5, 0.3, 0.2\}$, while those for group B are $\{0.9, 0.9, 0.9, 0.2\}$. Although group B exhibits a significantly better overall performance, Robustness@k assigns the same score to both groups.

where $MRR$ is mean reciprocal rank, $R_{og}$ is the rank of the document in the original retrieval mode, and $R_{new}$ is the new rank in the instruction mode. However, ③ $p$-MRR fails to distinguish the importance of ranking, neglecting to highlight the critical role that top K results play in retrieval. ④ Moreover, the linear discount mechanism of $p$-MRR is insufficiently sensitive to changes in higher rankings, making it ineffective in capturing subtle movements at the top. ⑤ Lastly, $p$-MRR demonstrates limitations when addressing special cases and extreme performances. [5]

## C    PROMPT FOR LIST-WISE RERANKING MODELS

Table 16: Prompt for List-wise Reranking Models. The input consists of a list of documents or passages, and the model is prompted to return a ranked list of document IDs based on their relevance to the query.

| TASK | Prompt Template |
|------|-----------------|
| Rank | <\|system\|>
You are RankGPT, an intelligent assistant that ranks passages based on their relevance to a query.
<\|user\|>
I will provide you with {num} passages, each indicated by a number identifier [ ]. Rank the passages based on their relevance to the query: {query}.

[1] {passage 1}
[2] {passage 2}
...
[num] {passage {num}}

Search Query: {query}.

Rank the {num} passages above based on their relevance to the search query. The passages should be listed in descending order using identifiers. The most relevant passages should be listed first. The output format should be [ ] >[ ], e.g., [1] >[2]. Only respond with the ranking results, do not say any word or explain.
<\|assistant\|>
*Model Generation:* [9] >[4] >[20] >... >[13] |

---

[5]For example, the performance of model 1 is $R_{og} = 10$ and $R_{new} = 5$, yielding a $p$-MRR of -0.5, while model 2's performance is $R_{og} = 100$ and $R_{new} = 50$, resulting in a $p$-MRR of -0.5. Although both models receive the same score, it is evident that model 1 has a greater impact on the retrieval results.

## D    SAMPLING OF WISE SCORE REWARD COMPONENT

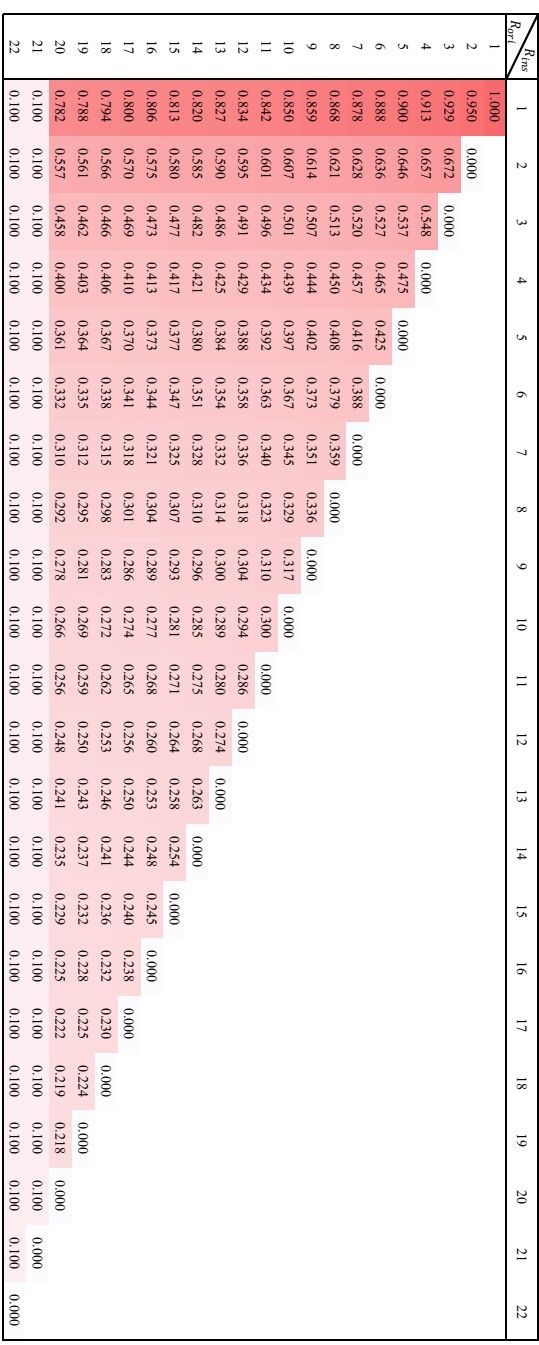

Figure 5: Heatmap of the rewards component

## E    THE COMPLETE RESULTS OF EVALUATING WITH INFOSEARCH DATASET

Table 17, Table 18, Table 19, Table 20, Table 21, and Table 22 show all the results of the 15 retrieval models in **InfoSearch** dataset. *Ori* indicates models evaluate in Original mode. *Ins* indicates models evaluate in Instructed mode. *Rev* indicates models evaluate in Reversely instructed mode. Act. indicates the actual performance of the model and ideal indicates the ideal performance. Per. indicates

how far the actual performance is from the ideal performance as a proportion of the ideal performance. A lower percentage indicates that the actual performance is closer to the ideal performance, while a higher percentage indicates a greater deviation from the ideal performance. The calculation formula is $Per. = \frac{ideal-actual}{ideal}$.

Table 17: Audience Results

| Model | nDCG@10 | | | MRR@1 | | | WISE | | | SICR ↑ |
|---|---|---|---|---|---|---|---|---|---|---|
| | Or | Ch | Re | Or | Ch | Re | Act. ↑ | Ideal ↑ | Per.↓ | |
| **Audience-(Layman, Expert)** | | | | | | | | | | |
| BM25 | 46.1 | 38.7 | 36.4 | 21.0 | 11.9 | 13.3 | -3.0 | 65.9 | 104.6 | 0.0 |
| Dense Retrieval | | | | | | | | | | |
| Bge-Large-v1.5 | 48.6 | 38.1 | 37.6 | 22.9 | 12.9 | 11.9 | -16.8 | 67.5 | 124.9 | 0.5 |
| E5-Large-v2 | 53.9 | 45.3 | 42.6 | 32.4 | 16.7 | 16.2 | -15.6 | 71.7 | 121.7 | 1.4 |
| Instructor-XL | 48.3 | 30.1 | 31.2 | 29.5 | 8.6 | 10.0 | -27.7 | 64.6 | 142.9 | 5.7 |
| Mistral-ins-v0.2 | 31.1 | 35.6 | 37.5 | 20.0 | 17.1 | 17.1 | -35.8 | 40.6 | 188.3 | 0.0 |
| E5-Mistral-ins | 78.9 | 63.3 | 64.3 | 72.4 | 34.8 | 35.2 | -7.3 | 86.1 | 108.5 | 2.9 |
| GritLM | 56.2 | 56.7 | 57.1 | 41.9 | 31.4 | 30.0 | -3.4 | 70.2 | 104.9 | 11.4 |
| GTE-Qwen2 | 56.4 | 57.0 | 57.3 | 46.7 | 35.2 | 35.2 | -34.0 | 65.3 | 152.0 | 0.0 |
| SFR-Embedding-2-R | 63.2 | 51.6 | 52.0 | 41.9 | 24.8 | 22.9 | -7.8 | 79.2 | 109.9 | 2.9 |
| NV-Embed-v2 | 65.3 | 47.6 | 47.5 | 44.8 | 17.6 | 17.1 | -9.8 | 80.5 | 112.2 | 2.4 |
| Point-wise Reranking | | | | | | | | | | |
| Mistral-ins-v0.2 | 75.8 | 60.9 | 63.6 | 62.9 | 28.1 | 35.2 | -8.9 | 85.0 | 110.4 | 1.4 |
| Llama-3.1 | 79.9 | 65.1 | 67.4 | 68.6 | 36.7 | 41.4 | -6.2 | 88.4 | 107.0 | 6.2 |
| FollowIR | 76.9 | 64.9 | 63.6 | 69.5 | 35.7 | 35.2 | -2.3 | 85.7 | 102.6 | 3.3 |
| List-wise Reranking | | | | | | | | | | |
| Mistral-ins-v0.2 | 68.7 | 58.9 | 58.6 | 51.4 | 29.0 | 28.6 | -6.3 | 81.0 | 107.8 | 10.5 |
| Zephyr-beta | 77.0 | 62.1 | 62.6 | 71.4 | 35.2 | 37.6 | -2.7 | 84.8 | 103.2 | 1.0 |
| RankVicuna-v1 | 62.2 | 52.5 | 51.2 | 50.5 | 27.1 | 25.7 | -2.5 | 75.1 | 103.3 | 5.2 |
| RankZephyr-v1 | 71.0 | 58.9 | 59.2 | 56.2 | 31.0 | 30.0 | 7.4 | 82.6 | 91.1 | 4.3 |
| GPT-4o | 87.7 | 72.5 | 72.6 | 88.6 | 48.6 | 48.6 | 7.4 | 95.9 | 92.2 | 15.2 |

Table 18: Keyword Results

| Keywords-(Include [keywords]) | | | | | | | | | | |
|---|---|---|---|---|---|---|---|---|---|---|
| Model | nDCG@10 | | | MRR@1 | | | WISE | | | SICR ↑ |
| | Or | Ch | Re | Or | Ch | Re | Act. ↑ | Ideal ↑ | Per.↓ | |
| BM25 | 70.4 | 70.0 | 54.1 | 64.5 | 45.3 | 24.7 | -42.1 | 77.7 | 154.2 | 0.0 |
| Dense Retrieval | | | | | | | | | | |
| Bge-Large-v1.5 | 46.2 | 39.7 | 29.4 | 25.8 | 11.8 | 11.5 | -38.2 | 66.2 | 157.8 | 0.0 |
| E5-Large-v2 | 60.1 | 70.6 | 45.0 | 43.2 | 46.7 | 16.0 | -38.3 | 75.6 | 150.7 | 0.7 |
| Instructor-XL | 68.5 | 48.7 | 38.4 | 56.8 | 19.9 | 14.6 | -34.7 | 79.4 | 143.7 | 2.1 |
| Mistral-ins-v0.2 | 31.7 | 28.5 | 37.0 | 30.7 | 7.3 | 19.9 | -67.8 | 36.0 | 288.4 | 0.0 |
| E5-Mistral-ins | 72.3 | 79.5 | 71.8 | 60.6 | 57.1 | 33.1 | -44.5 | 80.7 | 155.2 | 0.0 |
| GritLM | 85.9 | 79.4 | 67.2 | 89.2 | 58.2 | 46.0 | 6.8 | 86.6 | 92.2 | 11.8 |
| GTE-Qwen2 | 58.9 | 43.5 | 49.3 | 58.2 | 18.1 | 32.8 | -36.5 | 64.9 | 156.2 | 0.0 |
| SFR-Embedding-2-R | 47.3 | 64.5 | 47.7 | 30.7 | 38.3 | 24.4 | -45.9 | 65.4 | 170.3 | 1.0 |
| NV-Embed-v2 | 61.5 | 61.4 | 40.2 | 49.8 | 34.8 | 17.1 | -27.7 | 74.7 | 137.1 | 0.7 |
| Point-wise Reranking | | | | | | | | | | |
| Mistral-ins-v0.2 | 39.9 | 63.6 | 38.0 | 16.7 | 40.1 | 16.0 | 34.5 | 62.4 | 44.7 | 28.6 |
| Llama-3.1 | 61.7 | 76.9 | 48.4 | 48.4 | 54.0 | 28.2 | 38.7 | 74.2 | 47.8 | 38.3 |
| FollowIR | 51.2 | 78.1 | 45.7 | 34.1 | 59.9 | 25.8 | 47.7 | 68.7 | 30.6 | 27.2 |
| List-wise Reranking | | | | | | | | | | |
| Mistral-ins-v0.2 | 67.4 | 79.3 | 43.8 | 59.2 | 64.1 | 27.2 | 46.0 | 76.8 | 40.1 | 59.2 |
| Zephyr-beta | 68.9 | 65.2 | 47.8 | 59.9 | 47.0 | 32.8 | 14.1 | 77.0 | 81.7 | 27.5 |
| RankVicuna-v1 | 66.8 | 75.6 | 51.9 | 65.2 | 57.8 | 32.1 | -8.5 | 75.7 | 111.2 | 10.5 |
| RankZephyr-v1 | 72.6 | 77.3 | 52.0 | 79.1 | 60.6 | 34.5 | 53.9 | 92.3 | 41.6 | 42.5 |
| GPT-4o | 71.8 | 78.8 | 61.9 | 66.2 | 70.7 | 51.6 | 63.0 | 86.0 | 26.7 | 60.6 |

Table 19: Format Results

| Format-(Stackoverflow Post, Code Snippet, Official Manual | | | | | | | | | |
|---|---|---|---|---|---|---|---|---|---|
| Model | nDCG@10 | | | MRR@1 | | | WISE | | | SICR ↑ |
| | Or | Ch | Re | Or | Ch | Re | Act. ↑ | Ideal ↑ | Per.↓ | |
| BM25 | 22.6 | 15.3 | 19.1 | 16.0 | 4.7 | 8.3 | -2.8 | 30.7 | 109.0 | 0.0 |
| Dense Retrieval | | | | | | | | | | |
| Bge-Large-v1.5 | 58.0 | 25.9 | 31.4 | 46.0 | 5.3 | 11.3 | -42.1 | 65.8 | 163.9 | 0.3 |
| E5-Large-v2 | 59.3 | 44.9 | 52.0 | 44.0 | 20.3 | 36.7 | -15.5 | 68.9 | 122.5 | 0.0 |
| Instructor-XL | 64.2 | 35.7 | 40.6 | 54.0 | 11.7 | 20.3 | -30.5 | 72.5 | 142.1 | 0.3 |
| Mistral-ins-v0.2 | 2.4 | 3.2 | 4.3 | 0.0 | 1.0 | 1.3 | -29.7 | 5.6 | 630.7 | 0.0 |
| E5-Mistral-ins | 72.2 | 46.7 | 58.9 | 72.0 | 17.7 | 37.0 | -19.9 | 75.9 | 126.3 | 0.0 |
| GritLM | 45.5 | 48.4 | 53.8 | 31.0 | 21.3 | 38.7 | -36.0 | 54.6 | 165.9 | 1.3 |
| GTE-Qwen2 | 14.3 | 14.6 | 19.0 | 13.0 | 5.3 | 12.3 | -44.3 | 18.2 | 343.6 | 0.0 |
| SFR-Embedding-2-R | 75.4 | 53.0 | 63.1 | 76.0 | 23.7 | 46.3 | -13.0 | 78.5 | 116.5 | 2.0 |
| NV-Embed-v2 | 67.5 | 41.5 | 53.1 | 59.0 | 15.3 | 30.7 | -18.1 | 73.4 | 124.7 | 1.0 |
| Point-wise Reranking | | | | | | | | | | |
| Mistral-ins-v0.2 | 62.2 | 50.8 | 61.7 | 43.0 | 21.3 | 35.7 | -9.3 | 74.2 | 112.6 | 2.7 |
| Llama-3.1 | 68.0 | 51.2 | 59.5 | 58.0 | 18.3 | 32.7 | -9.5 | 77.3 | 112.3 | 10.0 |
| FollowIR | 72.2 | 54.9 | 68.5 | 62.0 | 23.3 | 50.0 | -2.3 | 79.7 | 102.9 | 7.0 |
| List-wise Reranking | | | | | | | | | | |
| Mistral-ins-v0.2 | 69.1 | 50.8 | 58.3 | 69.0 | 24.3 | 42.3 | -6.6 | 74.6 | 108.8 | 9.7 |
| Zephyr-beta | 48.8 | 35.2 | 42.3 | 51.0 | 15.3 | 36.7 | -13.9 | 58.3 | 123.8 | 8.0 |
| RankVicuna-v1 | 44.4 | 33.3 | 41.3 | 32.0 | 12.7 | 25.3 | -9.8 | 56.7 | 117.2 | 3.3 |
| RankZephyr-v1 | 73.0 | 52.6 | 63.5 | 61.0 | 20.3 | 41.7 | 7.8 | 81.2 | 90.4 | 1.0 |
| GPT-4o | 78.7 | 59.4 | 67.7 | 84.0 | 32.0 | 52.3 | 21.9 | 94.3 | 76.8 | 11.3 |

Table 20: Language Results

| Language-(Chinese, English) | | | | | | | | | | |
|---|---|---|---|---|---|---|---|---|---|---|
| Model | nDCG@10 | | | MRR@1 | | | WISE | | | SICR ↑ |
| | Or | Ch | Re | Or | Ch | Re | Act. ↑ | Ideal ↑ | Per.↓ | |
| BM25 | 36.1 | 30.3 | 28.8 | 28.0 | 14.0 | 13.0 | -7.2 | 43.7 | 116.5 | 0.0 |
| Dense Retrieval | | | | | | | | | | |
| Bge-Large-v1.5 | 42.7 | 36.1 | 32.0 | 38.0 | 19.0 | 13.5 | -20.7 | 51.5 | 140.1 | 2.0 |
| E5-Large-v2 | 52.4 | 50.2 | 44.6 | 58.0 | 35.5 | 33.5 | -25.3 | 56.7 | 144.7 | 0.5 |
| Instructor-XL | 47.6 | 37.7 | 35.0 | 54.0 | 25.0 | 19.5 | -20.5 | 50.0 | 141.1 | 4.0 |
| Mistral-ins-v0.2 | 20.7 | 30.0 | 29.5 | 19.0 | 21.0 | 19.0 | -31.9 | 26.5 | 220.6 | 0.0 |
| E5-Mistral-ins | 81.5 | 73.4 | 62.4 | 80.0 | 48.5 | 40.0 | 0.1 | 87.4 | 99.9 | 10.5 |
| GritLM | 82.6 | 81.0 | 75.9 | 78.0 | 57.0 | 48.0 | -6.7 | 87.7 | 107.7 | 4.5 |
| GTE-Qwen2 | 38.5 | 36.8 | 37.9 | 43.0 | 27.0 | 28.5 | -18.0 | 41.1 | 143.7 | 0.0 |
| SFR-Embedding-2-R | 81.5 | 81.7 | 64.3 | 77.0 | 61.0 | 31.5 | -15.4 | 88.1 | 117.4 | 1.0 |
| NV-Embed-v2 | 68.3 | 67.4 | 59.6 | 72.0 | 39.0 | 33.5 | -7.3 | 76.8 | 109.5 | 3.5 |
| Point-wise Reranking | | | | | | | | | | |
| Mistral-ins-v0.2 | 58.9 | 63.3 | 60.8 | 32.0 | 38.5 | 32.0 | 21.4 | 76.6 | 72.0 | 6.5 |
| Llama-3.1 | 67.9 | 71.9 | 64.2 | 61.0 | 44.5 | 36.5 | 29.0 | 79.4 | 63.5 | 22.0 |
| FollowIR | 68.4 | 70.6 | 64.7 | 54.0 | 48.0 | 37.0 | 20.9 | 81.2 | 74.3 | 19.5 |
| List-wise Reranking | | | | | | | | | | |
| Mistral-ins-v0.2 | 73.7 | 70.7 | 63.5 | 69.0 | 49.5 | 38.5 | 7.6 | 81.9 | 90.7 | 23.0 |
| Zephyr-beta | 70.9 | 58.7 | 58.1 | 68.0 | 38.5 | 37.5 | -6.9 | 79.3 | 108.7 | 10.5 |
| RankVicuna-v1 | 63.4 | 55.5 | 53.2 | 54.0 | 29.5 | 29.5 | -11.8 | 76.3 | 115.5 | 4.5 |
| RankZephyr-v1 | 79.5 | 66.6 | 66.9 | 69.5 | 38.5 | 37.5 | 10.6 | 88.0 | 87.9 | 5.5 |
| GPT-4o | 83.2 | 86.2 | 82.2 | 83.0 | 75.5 | 65.5 | 53.1 | 91.3 | 41.8 | 55.5 |

Table 21: Length Results

| Length-(Sentence, Paragraph, Article) | | | | | | | | | | |
|---|---|---|---|---|---|---|---|---|---|---|
| Model | nDCG@10 | | | MRR@1 | | | WISE | | | SICR ↑ |
| | Or | Ch | Re | Or | Ch | Re | Act. ↑ | Ideal ↑ | Per.↓ | |
| BM25 | 63.3 | 43.4 | 54.1 | 64.0 | 19.3 | 40.7 | -7.5 | 71.5 | 110.4 | 0.0 |
| Dense Retrieval | | | | | | | | | | |
| Bge-Large-v1.5 | 62.7 | 35.1 | 42.0 | 46.0 | 9.0 | 15.7 | -28.6 | 74.3 | 138.4 | 4.7 |
| E5-Large-v2 | 73.9 | 52.6 | 63.0 | 66.0 | 26.3 | 48.0 | -21.6 | 80.3 | 126.9 | 2.7 |
| Instructor-XL | 75.7 | 38.1 | 47.3 | 74.0 | 12.3 | 25.3 | -35.5 | 81.3 | 143.7 | 1.7 |
| Mistral-ins-v0.2 | 11.8 | 22.6 | 27.6 | 13.0 | 11.7 | 25.0 | -66.6 | 13.7 | 586.2 | 0.0 |
| E5-Mistral-ins | 86.2 | 64.2 | 76.2 | 92.0 | 33.0 | 60.3 | -13.4 | 86.0 | 115.6 | 0.0 |
| GritLM | 74.0 | 65.4 | 76.6 | 76.0 | 36.0 | 61.7 | -25.8 | 79.0 | 132.6 | 0.3 |
| GTE-Qwen2 | 34.4 | 47.3 | 55.0 | 32.0 | 22.3 | 42.7 | -56.4 | 40.7 | 238.6 | 0.3 |
| SFR-Embedding-2-R | 75.7 | 59.1 | 70.7 | 70.0 | 27.7 | 53.3 | -13.5 | 81.7 | 116.5 | 1.0 |
| NV-Embed-v2 | 81.9 | 55.3 | 65.6 | 83.0 | 26.7 | 48.0 | -9.3 | 84.6 | 111.0 | 0.3 |
| Point-wise Reranking | | | | | | | | | | |
| Mistral-ins-v0.2 | 60.3 | 52.3 | 60.2 | 42.0 | 24.0 | 38.0 | -12.6 | 74.4 | 116.9 | 4.7 |
| Llama-3.1 | 84.4 | 64.1 | 73.0 | 86.0 | 36.7 | 56.0 | -5.9 | 87.6 | 106.8 | 2.7 |
| FollowIR | 80.3 | 61.9 | 71.5 | 80.0 | 35.0 | 55.7 | -2.6 | 84.6 | 103.0 | 1.7 |
| List-wise Reranking | | | | | | | | | | |
| Mistral-ins-v0.2 | 88.7 | 66.0 | 76.8 | 92.0 | 38.7 | 64.7 | -1.9 | 89.1 | 102.2 | 8.0 |
| Zephyr-beta | 85.8 | 61.5 | 74.6 | 93.0 | 31.7 | 63.0 | -5.7 | 86.5 | 106.6 | 2.0 |
| RankVicuna-v1 | 84.8 | 61.8 | 74.1 | 93.0 | 32.7 | 60.7 | -4.3 | 85.4 | 105.0 | 2.3 |
| RankZephyr-v1 | 86.1 | 63.5 | 75.7 | 90.0 | 34.0 | 59.7 | 7.8 | 88.2 | 91.1 | 4.3 |
| GPT-4o | 89.1 | 68.8 | 79.0 | 95.0 | 42.0 | 67.3 | 10.2 | 94.4 | 89.2 | 10.3 |

Table 22: Source Results

| Model | Source-(Blog, Forum Post, News Article) | | | | | | | | | |
|-------|------|------|------|------|------|------|------|------|------|------|
| | nDCG@10 | | | MRR@1 | | | WISE | | | SICR ↑ |
| | Or | Ch | Re | Or | Ch | Re | Act. ↑ | Ideal ↑ | Per.↓ | |
| BM25 | 45.8 | 37.0 | 38.3 | 25.0 | 15.7 | 15.7 | -9.4 | 63.4 | 114.9 | 0.0 |
| Dense Retrieval | | | | | | | | | | |
| Bge-Large-v1.5 | 59.1 | 34.6 | 36.8 | 49.0 | 12.3 | 16.7 | -30.7 | 71.4 | 143.0 | 2.3 |
| E5-Large-v2 | 61.1 | 48.6 | 52.1 | 49.0 | 22.7 | 29.0 | -23.2 | 72.6 | 132.0 | 1.3 |
| Instructor-XL | 70.5 | 40.0 | 43.7 | 66.0 | 13.0 | 15.3 | -29.8 | 77.7 | 138.3 | 4.0 |
| Mistral-ins-v0.2 | 19.9 | 32.9 | 39.2 | 18.0 | 12.0 | 24.0 | -63.3 | 23.9 | 364.7 | 0.0 |
| E5-Mistral-ins | 76.2 | 58.9 | 62.4 | 70.0 | 30.7 | 36.0 | -13.0 | 82.5 | 115.8 | 3.3 |
| GritLM | 80.3 | 66.3 | 67.5 | 76.0 | 40.7 | 46.0 | -1.5 | 84.4 | 101.8 | 11.7 |
| GTE-Qwen2 | 58.2 | 59.2 | 72.4 | 55.0 | 29.0 | 53.0 | -44.6 | 63.6 | 170.1 | 0.0 |
| SFR-Embedding-2-R | 78.9 | 63.1 | 62.6 | 74.0 | 37.0 | 35.3 | -13.2 | 84.0 | 115.7 | 4.7 |
| NV-Embed-v2 | 71.3 | 53.8 | 47.5 | 67.0 | 29.7 | 23.3 | -8.7 | 78.5 | 111.1 | 9.0 |
| Point-wise Reranking | | | | | | | | | | |
| Mistral-ins-v0.2 | 71.5 | 59.7 | 69.7 | 56.0 | 24.3 | 48.0 | -0.3 | 81.0 | 100.4 | 4.7 |
| Llama-3.1 | 84.9 | 71.4 | 79.9 | 91.0 | 46.0 | 75.7 | 40.2 | 84.8 | 52.6 | 36.7 |
| FollowIR | 81.9 | 67.4 | 79.3 | 74.0 | 35.7 | 65.0 | 18.8 | 86.1 | 78.2 | 16.3 |
| List-wise Reranking | | | | | | | | | | |
| Mistral-ins-v0.2 | 77.7 | 60.8 | 68.6 | 78.0 | 34.3 | 57.0 | 10.0 | 81.0 | 87.6 | 21.7 |
| Zephyr-beta | 70.4 | 52.9 | 62.8 | 65.0 | 23.7 | 44.0 | -3.9 | 78.2 | 105.0 | 3.0 |
| RankVicuna-v1 | 68.7 | 52.7 | 59.5 | 74.0 | 30.3 | 50.0 | -2.2 | 72.6 | 103.0 | 8.0 |
| RankZephyr-v1 | 82.8 | 61.8 | 71.0 | 85.0 | 33.7 | 56.0 | -0.3 | 83.5 | 100.4 | 5.3 |
| GPT-4o | 89.4 | 79.5 | 82.0 | 93.0 | 65.7 | 77.3 | 45.2 | 95.5 | 52.7 | 39.7 |