# OpenReview forum: "Beyond Content Relevance: Evaluating Instruction Following in Retrieval Models"
_ICLR.cc/2025/Conference — ICLR 2025 Poster_

### Official Review · Reviewer_7Ty8 · 2024-11-03

**Soundness:** 2
**Presentation:** 3
**Contribution:** 2
**Rating:** 3
**Confidence:** 4

**Summary:**

This paper introduces a new dataset for evaluating instructable retrievers. The dataset is based on existing IR data and utilizes GPT-4 to generate instructions across six dimensions. Additionally, it rewrites the original documents to align with the new instructions and includes reverse instructions. The paper proposes two new metrics to evaluate models’ sensitivity to instructions: (i) WISe, which rewards search results that comply with instructions and penalizes those that do not; and (ii) SICR, which considers both the relevance score predicted by the model and the ranking information. The paper evaluates different types of retrieval models, finding that the LLM-based listwise reranker performs the best, followed by the pointwise reranker and dense retriever.

**Strengths:**

- Compared to previous data, the new data considers more dimensions, providing better coverage for evaluation.

- This paper demonstrates the weaknesses of previous metrics for evaluating instruction following, such as Robustness@k and pMRR, and introduces two well-motivated alternative metrics.

- This paper conducts an extensive evaluation of different retrieval methods, with insightful results and analysis.

**Weaknesses:**

- The proposed data is incremental compared to existing data. For example, InstructIR includes the dimension of Audience, and this paper essentially reuses their data generation pipeline, adding more dimensions.

- Similar to InstructIR, both the instruction and the positive examples are generated by GPT-4. Although the authors mention they manually reviewed the quality, I am still concerned about the data’s reliability: the positive document is rewritten by GPT-4 to align with the instructions. However, GPT-4 itself serves as a baseline model with only moderate performance in determining whether a document follows an instruction.

- The proposed metric assumes there are n positive documents under the original mode, with only one remaining positive under the instruction mode. However, this assumption does not always hold in real-world tasks, which limits this metric’s applicability to their dataset alone. For instance, we might have multiple positives under the instruction mode or find that some positives are not strictly positive under the default mode.

**Questions:**

SICR uses relevance scores, but listwise models do not output relevance scores. How do you define the score for these models?

---

> ### Author Response · Authors · 2024-11-24
>
> > The proposed data is incremental compared to existing data. For example, InstructIR includes the dimension of Audience, and this paper essentially reuses their data generation pipeline, adding more dimensions.
>
> While there are some similarities between our data generation pipeline and that of InstructIR, there are also significant differences. InstructIR relies heavily on GPT-generated queries and documents. In contrast, our approach emphasizes preserving the naturalness of the dataset by selecting suitable Q-A pairs from existing datasets and then rewriting the documents to better align with our objectives. Additionally, our instructions are carefully designed to reflect real-world user scenarios, ensuring the dataset is more practical and applicable. These differences make our pipeline distinct and tailored to address specific needs.
>
> > Similar to InstructIR, both the instruction and the positive examples are generated by GPT-4. Although the authors mention they manually reviewed the quality, I am still concerned about the data’s reliability: the positive document is rewritten by GPT-4 to align with the instructions. However, GPT-4 itself serves as a baseline model with only moderate performance in determining whether a document follows an instruction.
>
> Thank you for raising this important point regarding the reliability of GPT-4-generated data and its impact on our experiments. While it is true that both the instructions and positive documents are rewritten by GPT-4, our pipeline incorporates additional steps to enhance the dataset's robustness. Specifically, beyond generating positive documents, we also create corresponding hard negatives using GPT-4, ensuring a challenging setup for downstream models. These hard negatives are designed to closely resemble the positive examples while deliberately introducing inconsistencies or irrelevant content, making the task of instruction-following evaluation more rigorous.
>
> Furthermore, to mitigate potential biases or limitations in GPT-4, we manually reviewed and curated the generated data to ensure alignment with the instructions and naturalness of the text.   This process provides an additional layer of quality assurance.
>
> > The proposed metric assumes there are n positive documents under the original mode, with only one remaining positive under the instruction mode. However, this assumption does not always hold in real-world tasks, which limits this metric’s applicability to their dataset alone. For instance, we might have multiple positives under the instruction mode or find that some positives are not strictly positive under the default mode.
>
> We acknowledge that the proposed metric assumes a specific setup where there is one positive document under the instruction-following mode, which may not always align with real-world tasks where multiple positives or varying definitions of positivity could exist. However, it is important to note that our dataset and metric are designed for offline evaluation, focusing specifically on instruction-following scenarios.
>
> Existing datasets and metrics primarily target semantic relevance, but there is currently no standardized benchmark for evaluating instruction-following capabilities. The goal of our work is to bridge this gap by introducing a dataset and metric tailored to this underexplored aspect.
>
> That said, we recognize the importance of extending our approach to better align with real-world tasks where instruction-following systems might operate in more dynamic settings. As part of our future work, we plan to explore how our metric and dataset can be adapted to accommodate such scenarios, including cases with multiple positives or nuanced definitions of relevance.
>
> > SICR uses relevance scores, but listwise models do not output relevance scores. How do you define the score for these models?
>
> Yes, the listwise models do not output relevance scores. So we assign relevance scores to each document based on the rank (e.g., rank 1: relevance score = 1/1; rank 2: relevance score = 1/2; rank 3 relevance score = 1/3, and so on). However, it does not matter how to assign different relevance scores to documents of different rank, as long as the larger rank, the smaller the relevance score. In essence, SICR is designed based on the rank.

---

> ### Author Response · Authors · 2024-11-30
>
> Dear Reviewer,
>
> Happy Thanksgiving! We sincerely appreciate your constructive feedback and the time you took to review our paper. We hope our responses to your comments have adequately addressed your concerns and questions. If you have any further feedback or additional points that you would like us to clarify, it would be extremely valuable for improving this study. We look forward to hearing from you.
>
> We look forward to your further feedback.
>
> Best regards,
>
> Author

---

> > ### Comment · Reviewer_7Ty8 · 2024-12-01
> >
> > Hi, Happy Thanksgiving to you as well!
> >
> > Thank you for your response! I am still not convinced by the claim that “the data reflects real-world user scenarios.” While the new data improves upon InstructIR by using more data sources, it is still artificially constructed and does not include instructions from real users.
> >
> > About the second point, although it’s good to see attempts to use LLMs to generate negative samples, I think the issue is not just “something has been done” but rather “whether things are executed as expected.” Currently, the guarantee relies on “having a manual review step,” but no details about this step are provided. For example, we need to understand the quality of the data annotated by LLMs, how much data has been filtered, and how consistent the results are. I am concerned about this because using LLMs to assess simple relevance remains a questions (e.g., https://arxiv.org/pdf/2411.08275). This paper attempts to use LLMs to label more (instructions, positive samples, and negative samples), but it does not provide sufficient validation.
> >
> > As for the metric, it’s great to see a new metric being introduced. However, the lack of generalization remains a weakness. Lastly, thank you for clarifying SICR.

---

> > > ### Author Response · Authors · 2024-12-03
> > >
> > > Dear Reviewer,
> > >
> > > As the discussion period deadline approaches, we would like to follow up to ensure that our clarifications and additional results address the concerns you raised.
> > >
> > > Regarding your concern that **"while the new data improves upon InstructIR by using more data sources, it is still artificially constructed and does not include instructions from real users,"** we sourced our data from diverse, well-established datasets, such as **MSMARCO (real user queries from Bing)**, **Stack Overflow**, and **Stack Exchange (real-world user questions)**, as well as domain-specific datasets like BioASQ and publichealth-qa. The queries and documents were carefully structured into six dimensions—Audience, Keyword, Format, Language, Length, and Source—ensuring systematic evaluation across various document attributes. **Unlike InstructIR, which pools heterogeneous queries, InfoSearch is deliberately structured to focus on specific, multi-faceted dimensions.** This design enables us to analyze models’ performance not only on semantic relevance but also on their ability to follow detailed, document-level instructions accurately and consistently, simulating real-world retrieval preferences.
> > >
> > > Regarding your concern about **validation**, we **implemented a robust multi-step process** to ensure the quality of synthetic data. We employed **Model-Assisted Coarse Validation** with 12 retrieval models (e.g., bge, e5-mistral, NV-Embed). Documents flagged for review based on ranking or relevance thresholds were manually verified. Similarly, **Annotator Validation** by graduate students with proven English proficiency achieved an inter-annotator agreement score of 92%, ensuring high annotation consistency. For **instruction checks**, instructions underwent manual review, achieving 99% accuracy for five dimensions, while Keyword instructions initially had a 3% error rate, which was iteratively corrected. A re-check confirmed that all instructions accurately matched their corresponding documents.
> > >
> > > If these updates resolve your concerns, would you mind adjusting your rating accordingly? Please let us know if there are any remaining points we should address.
> > >
> > > Thank you for your valuable feedback!
> > >
> > > Best regards,
> > >
> > > Author

---

> > > > ### Comment · Reviewer_7Ty8 · 2024-12-03
> > > >
> > > > Hi, Thank you for the detailed response!
> > > >
> > > > It’s good to see that diverse datasets are being used, but my main concern is that the dimensions being evaluated (Audience, Keyword, etc.) are artificially designed rather than derived from real user expressions. For example, in the Audience evaluation, queries from SciFact are used, which according to https://arxiv.org/pdf/2004.14974, are atomic verifiable statements. These queries are designed to be natural and inherently lack user background info. In this paper, to convert SciFact to instruction mode, the user background, and the positive and negative documents, are all generated by LLMs. I find this approach artificial and may not reflect real-world user information-seeking scenarios.  Other eval dimensions face similar issues.
> > > >
> > > > Therefore, while the new data is an improvement over InstructIR, the refinement appears incremental. And my concern about the "incremental improvement over existing data" remains.
> > > >
> > > > The additional details about data construction and human validation are very helpful. Given this info, I may consider the paper a weak reject.

---

> > > > > ### Author Response · Authors · 2024-12-03
> > > > >
> > > > > We respectfully but strongly disagree with your assertion that InfoSearch represents only an incremental improvement over InstructIR.
> > > > >
> > > > > **1. Fundamental Limitations of InstructIR in Measuring Instruction-Following:**
> > > > > - Our experimental results clearly demonstrate that **Robustness and nDCG**, the metrics heavily relied upon in InstructIR, are **not sensitive to changes in instructions**. For example, a model achieving high nDCG might fail to adapt rankings based on nuanced instruction changes, exposing the inadequacy of these metrics for evaluating instruction-following performance.
> > > > > - To address this critical gap, we introduce **two novel metrics—SICR and WISE**—that simultaneously consider instruction-mode and reverse-instruction-mode. These metrics **strictly and comprehensively measure** instruction-following performance by evaluating models’ ability to comply with both affirmative and negated constraints. This represents a **significant methodological advancement** and enables InfoSearch to assess dimensions of instruction-following that InstructIR cannot.
> > > > >
> > > > > **2. Systematic Coverage of Diverse Instruction Types:**
> > > > > - InfoSearch evaluates retrieval models across **six meta-dimensions—Audience, Keyword, Format, Language, Length, and Source**—to provide a comprehensive and systematic view of how retrieval models perform under diverse instructional scenarios. These dimensions allow researchers to pinpoint **specific areas of strength and weakness** for models.
> > > > > - InstructIR, despite its 9,906 queries, focuses solely on content alignment, lacking insight into **why a model succeeds or fails on certain instructions.** Its absence of clear categorization limits its ability to provide actionable insights into retrieval performance. In contrast, InfoSearch is **explicitly structured to identify** how models handle different types of user preferences and instructions, ensuring **a deeper and more practical understanding** of model behavior.
> > > > >
> > > > > **3. Beyond Content Relevance:**
> > > > > - A key limitation of InstructIR is that its instructions **remain document-content-focused**, even for complex cases. For instance, instructions such as: *“I prefer to invest more in experiences than in transport, so I need accurate information on the annual fee for Spirit Airlines’ membership or credit card programs that offer benefits like waived bag fees or discounts.”* still focus solely on retrieving content-related information.
> > > > > - In contrast, InfoSearch, as its title suggests—**“Beyond Content”—explicitly evaluates document-level attributes such as style, format, language, and length in addition to content**. For example, models are required to retrieve documents of a particular format (e.g., blog, code snippet, or official manual) or adapt to the user's preferred tone (e.g., layman-friendly or expert-level). This goes far beyond content relevance and introduces a more holistic evaluation of instruction-following abilities.
> > > > >
> > > > > **4. Real-World Relevance of InfoSearch:**
> > > > > - While we acknowledge that **no dataset perfectly mirrors real-world user experience, a limitation shared by all existing benchmarks**, including InstructIR, InfoSearch bridges this gap by using core query/document pairs from **MSMARCO**, **StackOverflow** and so on, datasets that represent real-world information requests. This ensures a foundation rooted in practical and common search scenarios.
> > > > > - Beyond real-world relevance, our primary goal is to provide **a benchmark for evaluating models' instruction-following capabilities**, especially as many models claim proficiency in this area. Our six meta-dimensions are derived from **fundamental information needs**—such as cross-language retrieval, keyword-based constraints, or audience adaptation. **The poor performance of models on these dimensions**, as revealed by our results, highlights significant gaps and **charts clear directions for future improvements.**
> > > > >
> > > > > We welcome **specific and constructive suggestions** to further improve InfoSearch and are committed to advancing the field by providing a meaningful benchmark that benefits the broader community.

---

> > > > > > ### Comment · Reviewer_7Ty8 · 2024-12-03
> > > > > >
> > > > > > I agree that InfoSearch covers more dimensions and data sources compared to InstructIR.
> > > > > >
> > > > > > I think it is an incremental improvement over InstructIR because it reuses the InstructIR data generation pipeline and also use an LLM (GPT-4) to generate instructions, positives, and negatives. I found the data generation pipeline to be somewhat incremental, with the quality and realism of the data being major concerns. Specifically, GPT-4 itself serves as a baseline and only correctly identify which docs follow the instruction 33.5% of time.
> > > > > >
> > > > > > *My overall impression of this pipeline is that for a given question, GPT-4 generates an instruction, then generates a positive doc that follows it and some negative docs that do not. In evaluation, GPT-4 is a baseline to determine which doc follows the instruction, getting only 33.5% accuracy. This process seems too artificial. Correct me if I have fundamentally misunderstood anything.*
> > > > > >
> > > > > > The newly provided details about human validation seem to address these quality concerns to some extent, but I am still not entirely convinced (e.g., the pipeline, the example of SciFact).
> > > > > >
> > > > > > In general, my main concern is the reliance on GPT to generate test data. The original paper lacks validation of the reliability of LLM-generated data.
> > > > > >
> > > > > > Lastly, I would like to reference two recently published papers that explore a same idea of evaluating diverse instruction types but use real data instead. I think this paper could add discussion in future version: https://arxiv.org/pdf/2405.02714 (at COLM-24, eval six dimensions that appear similar to this paper) and https://arxiv.org/pdf/2410.10127 (at EMNLP-24, a collection of many IR tasks with instruction input).

---

> > > > > > > ### Author Response · Authors · 2024-12-03
> > > > > > >
> > > > > > > > I agree that InfoSearch covers more dimensions and data sources compared to InstructIR.
> > > > > > >
> > > > > > > The primary contribution of our work lies not in data generation and coverage alone but in the structured design and metrics for evaluating instruction-following. The six meta-dimensions of instructions provide systematic insights into model behavior, and the SICR and WISE metrics allow for strict and comprehensive evaluation. These elements enable researchers to assess instruction-following in ways that prior benchmarks like InstructIR cannot.
> > > > > > >
> > > > > > > We acknowledge that, as the first benchmark of its kind, InfoSearch has limitations in data coverage and realism. We are committed to addressing these limitations in future iterations. Specifically, we aim to include more diverse dimensions and categories of instructions, incorporating those from existing datasets like InstructIR, while exploring additional real-world data sources to enhance realism and coverage.
> > > > > > >
> > > > > > > > In evaluation, GPT-4 is a baseline to determine which doc follows the instruction, getting only 33.5% accuracy. This process seems too artificial.
> > > > > > >
> > > > > > > We appreciate your feedback and would like to clarify several points to address the potential misunderstanding:
> > > > > > >
> > > > > > > 1. **Clarification on "33.5%”**
> > > > > > >
> > > > > > > **The metric of "33.5%" does not reflect the quality of the data generated by GPT-4o.** Instead, **WISE** (a ranking-based metric) evaluates the **rank change** of the gold document (i.e., the ground truth document) before and after incorporating instructions. Specifically:
> > > > > > >
> > > > > > > - The metric measures how well a system ranks the gold document when instructions are present, reflecting the effectiveness of instruction-following.
> > > > > > > - For GPT-4o, the performance is **quantified by R_{ins}=1.7** (as shown in Table 2), meaning GPT-4o ranks the gold document at an average position of 1.7—demonstrating strong performance.
> > > > > > > 2. **How We Use GPT-4 to Generate Instructions and Rewrite Documents**
> > > > > > >
> > > > > > > **Sec 1 Genreate Instructions**
> > > > > > >
> > > > > > > The **Length** dimension represents a user's potential preference for the length of retrieved answers. Before generating instructions, we manually define the **conditions**: `[sentence]`, `[paragraph]`, and `[article]`. For example:
> > > > > > >
> > > > > > > - **Query**: *How many calories are in a martini?*
> > > > > > > - **Condition**: `[sentence]`
> > > > > > >
> > > > > > > The prompt is designed to guide the LLM to generate a query with an instruction (due to word limit constraints, only an outline of the prompt is provided. More details in appendix).
> > > > > > >
> > > > > > > ```
> > > > > > >  ### TASK
> > > > > > >  You are tasked with .... You will be provided with a query and a condition and you need to:
> > > > > > >  1. Rephrase the core query as the first sentence, .....
> > > > > > >  2. Create a second sentence that specifies the search restriction ....
> > > > > > >  3. Ensure each sentence is smooth ....
> > > > > > >
> > > > > > >  ### INPUT
> > > > > > >  Core Query: {core query}
> > > > > > >  Condition: {condition}
> > > > > > >
> > > > > > >  ### FORMATTING
> > > > > > >  your output should be JSON and in the following format:
> > > > > > > ....
> > > > > > > ```
> > > > > > >
> > > > > > > Results:
> > > > > > >
> > > > > > > - **Query with Instruction**: *How many calories are in a martini? Please give me a sentence answer.*
> > > > > > >
> > > > > > > And we generate reversely instruction:
> > > > > > >
> > > > > > > - **Query with Reversely Instruction:**  *How many calories are in a martini? Please provide a detailed response, not just a single sentence.*
> > > > > > >
> > > > > > > For different conditions, we expand the expression of the conditions to ensure diversity in phrasing.
> > > > > > >
> > > > > > > **Sec2 Rewrite Documents**
> > > > > > >
> > > > > > > For documents, the QA pairs in our collected dataset often satisfy only one specific condition. To address this, we rewrite the documents to align with the target condition while preserving the original content.
> > > > > > >
> > > > > > > For example:
> > > > > > >
> > > > > > > - **Query**: *What is the outlook for Canavan Disease?*
> > > > > > > - **Original Document**: *The prognosis for Canavan disease is poor. Death usually occurs before age 10, although some children may survive into their teens and twenties.….*
> > > > > > >     - This satisfies the `[paragraph]` condition.
> > > > > > >
> > > > > > > To create a document matching the `[sentence]` condition, we format the query, document, and target condition into a prompt and use the LLM for rewriting.
> > > > > > >
> > > > > > > ```
> > > > > > >  ### TASK
> > > > > > >  You are tasked with .... and you need to:
> > > > > > >  1. Create a concise sentence format.....
> > > > > > >  2. Create an article format of the document that expands and elaborates .....
> > > > > > >  3. Ensure both outputs are smooth ....
> > > > > > >
> > > > > > >  ### CAUTION
> > > > > > >  1. Do not change the factual content or ....
> > > > > > >  2. The sentence should be short and focused, while the article should provide sufficient elaboration and context.
> > > > > > >
> > > > > > >  ### INPUT
> > > > > > >  Core Query: {core query}
> > > > > > >  Document: {document}
> > > > > > >
> > > > > > >  ### FORMATTING
> > > > > > >  Your output should be JSON and in the following format:
> > > > > > > ....
> > > > > > > ```
> > > > > > >
> > > > > > > The result is:
> > > > > > >
> > > > > > > - **Rewritten Document (Sentence)**: *The outlook for Canavan Disease is generally poor.*
> > > > > > > - **Rewritten Document (Article)**:  *Canavan Disease, also known as Canavan-Van Bogaert-Bertrand disease, is a rare .... /n The prognosis for Canavan disease is generally poor, with death usually occurring before age 10 .... /n ....*
> > > > > > >
> > > > > > > This ensures that the rewritten documents align with the specified condition and the corresponding instructions.

---

> > > > > > > ### Author Response · Authors · 2024-12-03
> > > > > > >
> > > > > > > > Lastly, I would like to reference two recently published papers that explore a same idea of evaluating diverse instruction types but use real data instead.
> > > > > > >
> > > > > > > Thank you for providing the links to the two papers. We have carefully reviewed them and gained valuable insights. In our future work, we will reference and integrate the perspectives from Beyond Relevance into our research framework. Additionally, we plan to further expand the design of our dataset, similar to MAIR, by incorporating statistical analysis of the distribution of existing IR datasets at the "dimension" level in this study, which will enhance the dataset’s comprehensiveness and practical utility. Once again, we greatly appreciate your valuable suggestions and support!

---

> ### Author Response · Authors · 2024-12-01
>
> Thank you for your response! We appreciate your insightful comments and will address each of your concerns in detail below.
>
> > “the data reflects real-world user scenarios.” While the new data improves upon InstructIR by using more data sources, it is still artificially constructed and does not include instructions from real users.
>
> We sourced our data from several well-established and diverse datasets to ensure broad coverage.The queries in **MSMARCO** are sourced from real user searches on **Bing**, while datasets like **Stack Overflow** and **Stack Exchange** also consist of questions posed by real users in real-world scenarios. Here are existing datasets we used.
> - Audience:  BioASQ, scifact
> - Keyword: MSMARCO
> - Format:  Stackoverflow, Stack Exchange
> - Language:  publichealth-qa
> - Length:  medical qa
> - Source:  CNN-english-news
>
> While we recognize the overlap in intent, we want to clarify the distinct purpose and methodology behind InfoSearch.  Unlike InstructIR, which pools heterogeneous queries together, our approach is **deliberately structured** around six distinct dimensions: Audience, Keyword, Format, Language, Length, and Source.  This structured design allows us to systematically evaluate retrieval models by analyzing not only the dimensions where they excel but also their ability to follow specific instructions accurately and consistently.
>
> The most significant distinction lies in InfoSearch's focus on structured, multi-faceted evaluation.  Each dimension represents a concrete document attribute, and we assess models across original, instructed, and reversely instructed modes, simulating real-world user preferences.  This approach enables us to explore instruction-following capabilities beyond content relevance by incorporating document-level attributes like format and language, which are often overlooked in prior work.
>
> > This paper attempts to use LLMs to label more (instructions, positive samples, and negative samples), but it does not provide sufficient validation.
>
> **Document Construction**
>
> - Audience: The Q-A pairs we collected were originally designed for [expert]. Using LLMs, we rewrote these documents to align with the [layman] condition, simplifying technical jargon while maintaining content relevance.
> - Keyword: Over 90% of the documents were sourced directly from MSMARCO. For certain documents, we used LLMs to refine and rewrite them to better match the instructions.
> - Format: All positive documents were extracted from StackOverflow posts and linked resources. No LLM generation was involved here.
> - Language: Documents were directly sourced from publichealth-qa.
> - Length: [paragraph] documents were derived from medical_qa. LLMs were employed to rewrite these documents to fit [sentence] and [article] conditions.
> - Source: [news] documents were obtained from CNN-english-news, and LLMs were used to transform them into [blog] or [forum] formats.
>
> **Hard Negative Documents**
>
> All hard negative documents were generated by LLMs to appear contextually similar to the query but deliberately irrelevant.
>
> **Instruction Construction**
>
> LLMs were used to expand predefined conditions (e.g., [sentence], [paragraph], [article]) into diverse, user-friendly instructions.
> To validate the quality of the dataset, we implemented the following multi-step process:
>
> 1. Model-Assisted Coarse Validation
>
> We used **12 retrieval models** (e.g., bge, e5-mistral, NV-Embed) to perform a coarse-grained validation of the generated positive documents. Documents were flagged for review if they met either of the following criteria:
> - Ranked below 50 by six or more models.
> - Received a relevance score below 0.5 from six or more models.
> - These flagged documents underwent a further manual review to confirm whether they were appropriate as positive samples for the associated queries.
>
> We applied the same model-assisted and manual review process to hard negative documents. This ensured that the documents appeared contextually similar to the query while being definitively irrelevant, maintaining the robustness of our dataset.
>
> 2. Annotator Validation
>
> The annotations were reviewed by graduate students specializing in computer science, all of whom passed an English proficiency test to ensure accurate task comprehension. We achieved an inter-annotator agreement score of 92%, reflecting high consistency among annotators.
>
> 3. Instruction Check
> - For five dimensions (excluding Keyword), instructions were generated based on predefined conditions and underwent manual review. Due to their straightforward expansion process, the accuracy of these instructions after review was 99%.
> - For the Keyword dimension, instructions were generated by extracting document keywords, with an error rate of 3% in a sample of 20 queries. Errors were iteratively corrected using LLMs, and a recent re-check confirmed that all positive documents contain the specified keywords after adding instructions.

---

### Official Review · Reviewer_S77t · 2024-11-05

**Soundness:** 2
**Presentation:** 3
**Contribution:** 2
**Rating:** 5
**Confidence:** 4

**Summary:**

In this paper, the authors evaluate retrieval models' instruction-following capabilities and develop the InfoSearch benchmark covering several dimensions of documents such as audience, keyword, format, language, length, and source. They also introduce new metrics to assess the models’ responsiveness to instructions.

Pros:
- The proposed dataset is somewhat useful to the research community.
- The exploration to document attributes for instruction-aware retrieval is interesting.
- The experimental study is insightful.

Cons:
- It would be great if the authors could introduce more on the motivation and application scenario for instruction following retrieval. Based on my understanding, if the instruction following functionality over document-level features can be implemented by introducing LLMs for understanding the query intent then rewriting the query with customized filters.  There is no need for an advanced retrieval model.
Again, some examples might help reviewers understand the problem of instruction following retrieval and confirm whether this is a true research problem.
- Focusing only on the document-level instructions is not enough for evaluating instruction-followed retrieval systems.  Even for document-level instructions, some additional dimensions such as temporal and location, should also be considered.
- For the experiments, I would expect to add some customized filters (such as language requirement) on existing retrieval models.
- It is unfair to compare retrieval models with reranking models.  I think the most challenging part is the retrieval part, which may fail to retrieve relevant results without good instruction understanding.

**Strengths:**

Pros:
- The proposed dataset is somewhat useful to the research community.
- The exploration to document attributes for instruction-aware retrieval is interesting.
- The experimental study is insightful.

**Weaknesses:**

Cons:
- It would be great if the authors could introduce more on the motivation and application scenario for instruction following retrieval. Based on my understanding, if the instruction following functionality over document-level features can be implemented by introducing LLMs for understanding the query intent then rewriting the query with customized filters.  There is no need for an advanced retrieval model.
Again, some examples might help reviewers understand the problem of instruction following retrieval and confirm whether this is a true research problem.
- Focusing only on the document-level instructions is not enough for evaluating instruction-followed retrieval systems.  Even for document-level instructions, some additional dimensions such as temporal and location, should also be considered.
- For the experiments, I would expect to add some customized filters (such as language requirement) on existing retrieval models.
- It is unfair to compare retrieval models with reranking models.  I think the most challenging part is the retrieval part, which may fail to retrieve relevant results without good instruction understanding.

**Questions:**

- What is the underlying retrieval model for the reranking models in Table 3? (How the initial results are prepared?)

---

> ### Author Response · Authors · 2024-11-24
>
> > It would be great if the authors could introduce more on the motivation and application scenario for instruction following retrieval.
>
> We sincerely thank the reviewer for their insightful comments and suggestions.
>
> 1.Motivation and Application Scenario for Instruction-Following Retrieval
>
> Our proposed dimensions address the complexity of user instructions by influencing not just specific parts of the retrieved content but its overall structure and style. This requires retrieval models to achieve deep semantic understanding and contextual adaptation, going beyond traditional relevance-based approaches to better align with nuanced user needs.
>
> For instance, when a user specifies "Source" as news articles, they are interested in more than just neutral facts—they seek to understand how the media frames or evaluates a particular topic. This involves the biases, perspectives, and storytelling styles unique to news reporting. The user's assessment of the retrieved content will be influenced by the logical structure, language style, and headline formatting typical of journalism. By accurately interpreting and prioritizing the "Source" dimension, the retrieval model can provide more authentic and need-specific information that closely aligns with the user's intent. This not only enhances the effectiveness of the retrieval process but also pushes the model beyond semantic understanding toward a deeper comprehension of user needs.
>
> 2.Necessity of Advanced Retrieval Models
>
> Although query rewriting with filters can achieve some level of customization, it often falls short when faced with complex, multidimensional instructions. For example:
> When length constraints are specified (e.g., "a brief summary of the history of machine learning" vs. "a detailed timeline"), the model needs to understand the granularity of the content within the retrieval process itself, beyond what query rewriting can achieve.
>
> This example highlights the necessity of advanced retrieval models capable of embedding instruction-following behavior directly into their architecture, ensuring the seamless handling of diverse and nuanced requirements.
> In summary, our response highlights the importance of addressing multidimensional instructions in retrieval tasks and the limitations of existing approaches. By proposing advanced retrieval models and focusing on nuanced user needs, we aim to bridge the gap between query understanding and truly instruction-aware retrieval systems.
>
> > Focusing only on the document-level instructions is not enough for evaluating instruction-followed retrieval systems. Even for document-level instructions, some additional dimensions such as temporal and location, should also be considered.
>
> Thank you for pointing out this important aspect. We agree that focusing solely on document-level instructions is not sufficient for a comprehensive evaluation of instruction-following retrieval systems. While our current benchmark serves as a foundational step, we acknowledge the value of incorporating additional dimensions, such as temporal ,location-specific and other aspects. Expanding the benchmark to include these and potentially other dimensions is a key part of our future work, and we are committed to continuously improving its coverage and utility.
>
> > For the experiments, I would expect to add some customized filters (such as language requirement) on existing retrieval models.
>
> We acknowledge the importance of incorporating customized filters (such as language requirements) into existing retrieval models. Such extensions would enhance the applicability and practicality of the models. As our current work primarily focuses on validating the core model and evaluating its performance, these customized features were not included in the experiments. However, we will seriously consider integrating these filters into our experimental design in future work.
>
> > It is unfair to compare retrieval models with reranking models. I think the most challenging part is the retrieval part, which may fail to retrieve relevant results without good instruction understanding.
>
> Thank you for your valuable feedback.  We would like to clarify that this paper focuses on **offline evaluation** rather than online retrieval.  Specifically, we address the challenge by setting carefully designed instructions and reverse instructions for each query, ensuring that the evaluation is conducted in a controlled and consistent manner.  This setup allows us to isolate and analyze the effectiveness of the methods in reranking without the variability introduced by online retrieval performance.
>
> > What is the underlying retrieval model for the reranking models in Table 3? (How the initial results are prepared?)
>
> For reranking models, we use the **top 100 results from E5-mistral** for reranking.

---

> > ### Comment · Reviewer_S77t · 2024-11-26
> >
> > Thank the authors for the detailed reply. My main concern about the limitation of the dataset still exists, so I will keep my score unchanged.

---

> > > ### Author Response · Authors · 2024-12-03
> > >
> > > Dear Reviewer,
> > >
> > > As we approach the end of the discussion period, we would like to follow up to ensure that our latest clarifications and insights address your concerns.
> > >
> > > Regarding your request to elaborate on **the motivation and application scenarios for instruction-following retrieval**, our primary aim is to **bridge the gap between user expectations for retrieval systems and their current limitations**. Unlike LLMs, which excel at generating content based on nuanced instructions, **retrieval models often prioritize content relevance alone, failing to accommodate user-specific preferences like document format or target audience**. By introducing instruction-following capabilities, we enable retrieval systems to better align with user-defined constraints, enhancing tasks like **retrieval-augmented generation** in customer support or domain-specific knowledge synthesis. For example, pre-filtering expert-level content (e.g., scientific papers) reduces the adaptation burden on LLMs, resulting in more efficient and accurate outputs.
> > >
> > > Regarding the comparison of **retrieval and reranking models**, we understand your concern about the inherent differences in their roles and challenges.  Our evaluation emphasizes retrieval systems’ ability to handle instructions effectively, as this is often a bottleneck in achieving instruction compliance.  Nonetheless, we included reranking models for completeness and to provide a broader perspective on how models at different stages handle instruction-following tasks. A specific example of an e5-mistral search result is given below:
> > >
> > > Q: how many calories are in a martini
> > > |              | **sentence doc** | **paragraph doc** | **article doc** |
> > > |:-----------------:|:----------------:|:-----------------:|:---------------:|
> > > | Q                 | **90.75**            | 89.42             | 84.85           |
> > > | Q+[sentence]      | **84.88**            | 84.01             | 78.58           |
> > > | Q+[paragraph]     | 84.83            | **85.50**             | 79.54           |
> > > | Q+[article]       | **85.76**            | 84.95             | 80.49           |
> > > | Q+[non-sentence]  | **86.51**            | 85.15             | 81.31           |
> > > | Q+[non-paragraph] | **86.57**            | 84.87             | 81.07           |
> > > | Q+[non-article]   | **86.82**            | 85.33             | 81.76           |
> > >
> > > It is evident that for retrieval models, the most challenging part is the retrieval process itself, which may fail to retrieve relevant results without a good understanding of instructions.
> > >
> > > We hope these responses clarify our motivation and approach. If they address your concerns, we kindly ask if you would consider adjusting your rating accordingly. Please don’t hesitate to let us know if further clarification is needed.
> > >
> > > Thank you for your valuable feedback!
> > >
> > > Best regards,
> > >
> > > Author

---

> ### Author Response · Authors · 2024-11-30
>
> Thank you for your thoughtful feedback.  We appreciate your continued consideration of our work.  Regarding your concern about the dataset limitations, we would like to offer the following responses:
> 1.  This study primarily aims to fill the gap in benchmarks for evaluating instruction-following.  Document-level instructions are particularly effective due to their clarity and minimal ambiguity, making them ideal for this purpose.
> 2.  While we acknowledge the absence of non-document-level instructions, we believe this does not diminish the completeness or value of this study. Our benchmark fills an important need by offering a unified platform for instruction-following evaluation, which can be extended in future iterations. That said, we would greatly appreciate any specific suggestions or examples of non-document-level instructions you have in mind. Broadening the scope to include such instructions is a natural direction for future development.
> 3.  We also value your suggestion to include dimensions like temporal or location-specific attributes.  These attributes are general and less ambiguous, and align well with our goal of creating a practical and robust benchmark. We will explore these ideas in future updates to enhance the framework.
>
> Thank you again for your insightful comments and guidance, which will undoubtedly help us improve this work further.

---

### Official Review · Reviewer_RSj2 · 2024-11-05

**Soundness:** 3
**Presentation:** 2
**Contribution:** 3
**Rating:** 6
**Confidence:** 4

**Summary:**

This paper introduce InfoSearch, a novel benchmark aims to evaluate retriever models' instruction following capabilities. The authors find that existing metrics like nDCG, Robustness@k, and pMRR are not sufficient for evaluating instruction following capabilities. To more rigorously evaluate this capability, this paper introduce 2 WISe and SICR. This paper conduct extensive analysis regarding different ranking method.

**Strengths:**

1. This paper bridge the gap that no benchmarks are evaluating retriever models' instruction following capabilities
2. This paper introduces 2 metrics for rigorously evaluating instruction following capabilities
3. Extensive experimentations
4. Interesting interpretation and analysis of the main result
5. reverse mode is interesting!

**Weaknesses:**

1. Sec 2.3 Evaluation Metric, the motivation for designing WISe and SICR is not clear. In L181 - L183, weakness of Robustness@k is not clearly explained.

2. Sec 2.3 Evaluation Metric, since you are comparing with your metrics with pMRR and Robustness@k, it's good to have a comparison table showing your metric is better in handling cases that pMRR and Robustness@k fail, in order to convince your audience that your metrics are effective. Additionally, it would be helpful if you could compare and explain the difference between WISe and SICR.

^ 2 follow-up, I saw Table 2 comparing all metrics, I think it might be good to have a table listing all strengths/weakness at the first place.

3. In eq (1), $F(q) = f_{\textit{penalty}}(R_{ori}, R_{ins})$ if $\lnot R_{ins} < R_{ori} < R_{rev}$; whereas, eq (3) only gives $R_{ori} < R_{ins}$. More clarifications will be helpful.

4. One of my major concern is $\textbf{you do not explain anything about the dataset construction pipeline}$, which I will treat the core of this paper. For example, you lack details

    4.1 How you select web pages?

    4.2 Prompt for step 2? No reference at all. How can you check the quality?

    4.3 Quality check for all GPT-generated responses. There's no details. If they are in the appendix please reference them in the main text.

    4.4 Only saying $\textbf{manual review}$ is NOT acceptable. Biographies of annotators? Inter-annotator agreement score?

**Questions:**

1. One minor issue but it's prevalent throughout the paper: consider using ~\citep rather than \cite for in-text citation, e.g., L145

2. L161, where's your table 2.2?

3. L198, $\textbf{P}^i$ is unclear, what $i$ refers to?

4. L203, 2 periods

5. eq(2), why you select 20 and 0.01, how you decide 2K? Treat it as normalization factor?

6. L310-L311, for reranking model, can you explain more how it works?

7. It might be super helpful if you can include your benchmark example in the main text.

8. Do you consider using general-purpose LLM for doing point-wise reranking? Like llama-3.2, mixtral, qwen2, etc.?

---

> ### Author Response · Authors · 2024-11-24
>
> > Sec 2.3 Evaluation Metric, the motivation for designing WISe and SICR is not clear. In L181 - L183, weakness of Robustness@k is not clearly explained.
>
> In our revised version paper, we explain it. The Robustness@k metric oversimplifies the evaluation of a model's ability to follow instructions.
> 1. Even if a model demonstrates strong performance across the majority of queries, a single anomalously low score can reduce the overall robustness score.
>
> 2. Furthermore, focusing solely on the lowest score disregards variations in the model's responses to different instructions, thus failing to capture the overall performance trend. For instance, the nDCG@k scores for group A are {0.8, 0.5, 0.3, 0.2}, while those for group B are {0.9, 0.9, 0.9, 0.2}. Although group B exhibits a significantly better overall performance, Robustness@k assigns the same score to both groups.
>
> > Sec 2.3 Evaluation Metric, since you are comparing with your metrics with pMRR and Robustness@k, it's good to have a comparison table showing your metric is better in handling cases that pMRR and Robustness@k fail, in order to convince your audience that your metrics are effective.
>
> In our revised version paper, we explain it. P-MRR fails to distinguish the importance of ranking, neglecting to highlight the critical role that top K results play in retrieval.
> 1. Moreover, the linear discount mechanism of P-MRR is insufficiently sensitive to changes in higher rankings, making it ineffective in capturing subtle movements at the top.
>
> 2. Lastly, P-MRR demonstrates limitations when addressing special cases and extreme performances. For example, the performance of model 1 is R_{og} = 10 and R_{new} = 5, yielding a **P-MRR of -0.5**, while model 2’s performance is R_{og} = 100 and R_{new} = 50, resulting in a **P-MRR of -0.5**. Although both models receive the same score, it is evident that model 1 has a greater impact on the retrieval results. And in the appendix, we We provide a heatmap.
>
> > In eq (1), F(q)=fpenalty(Rori,Rins) if ¬Rins<Rori<Rrev; whereas, eq (3) only gives Rori<Rins. More clarifications will be helpful.
>
> In our revised version of the paper, we have added further clarifications to address this concern. Additionally, we have addressed other issues in the paper, such as corrections to Table 2.2 for clarity and accuracy, and resolving citation formatting errors (e.g., \cite) to align with standard practices.
>
> > How you select web pages?
>
> To select web pages, we started with the original Q-A pairs from existing datasets. For example, in the "Source" dimension, we used the CNN-English-News dataset, where the documents only satisfied the [news article] condition. To create the [forum post] condition, we searched for related answers on platforms such as Reddit and Quora using the same query. For the [blog] condition, we performed targeted searches on Google to find relevant blog posts.
>
> > How can you check the quality?
>
> For GPT-generated documents, we follow a clear and structured guideline to ensure quality:
>
> 1. Positive Documents: We verify whether the generated document can effectively answer the query. Additionally, we check if it also satisfies the given instruction while maintaining relevance to the query.
>
> 2. Hard Negative Documents: We assess whether the generated document has a high semantic relevance to the query but fails to provide an answer. This ensures that the document poses a meaningful challenge for retrieval models.
>
> > Pi is unclear, what i refers to?
>
> Pi refers to the specific gold document within the set of positive documents P associated with a query q. This document becomes the focus in the instructed mode, where a more specific instruction highlights it as the only relevant document. In contrast, the other documents in P are considered irrelevant or negative under this mode.
>
> > for reranking model, can you explain more how it works?
>
> For reranking models, we use the **top 100 results from E5-mistral** for reranking. For point-wise setting, the model evaluates each query-document pair independently. Both the query and a single document are input into the model, and it outputs the probability of the document being relevant to the query. These probabilities are then used as similarity scores for ranking. For example, the model might determine the likelihood of "True" or "False" for each pair, which serves as the basis for re-ranking. For list-wise setting, this approach involves considering the entire list of candidate documents simultaneously.  The list is provided as a prompt, often including the query and all candidate documents.  The model processes the list and directly outputs the ranked order of document IDs.
>
> > Do you consider using general-purpose LLM for doing point-wise reranking? Like llama-3.2, mixtral, qwen2, etc.?
>
> We use LLama-3.1 for doing point-wise reranking and in our future work, we plan to explore incorporating general-purpose large language models (LLMs) such as Llama-3.2, Mixtral, and Qwen2.

---

> ### Comment · Reviewer_RSj2 · 2024-11-26
>
> Did you change the default submission format, _i.e.,_ remove the anonymous author lines? I understand you want to save more spaces but I'm not sure if this is acceptable.
>
> Thank you for explaining my questions regarding Sec 2.3. However, my main concern is the quality check (also pointed out by Reviewer 7Ty8). Who are the annotators? Any annotator biographies? Any inter-annotator agreement scores? Relying solely on GPT-4 might not be reliable.
>
> After reading reviews from other reviewers, as pointed out by Reviewer S77t, it would be beneficial to explain more on:
> - The overall motivation of the work -- instruction-following information retrieval and possible applications
> - Document-level instructions might overlook some "detailed" features within documents

---

> ### Author Response · Authors · 2024-11-30
>
> > Who are the annotators? Any annotator biographies? Any inter-annotator agreement scores?
>
> The annotators are graduate students specializing in computer science who were carefully selected and trained for this task. Each annotator underwent an English proficiency test to ensure they could accurately understand and evaluate the documents. The inter-annotator agreement score, calculated based on their judgments of whether a document is a positive or negative match for a given query, is **92%**. This high agreement reflects the consistency and reliability of the annotations.
>
> > The overall motivation of the work -- instruction-following information retrieval and possible applications
>
> The primary motivation of this work is to address the **gap between user expectations for instruction-following capabilities in retrieval models and their current limitations**. While large language models (LLMs) excel at generating content based on detailed instructions, retrieval systems have lagged behind, focusing primarily on content relevance without adequately considering user-specific instructions. This limits their applicability in scenarios where nuanced preferences, such as document format or audience, are critical.
>
> Instruction-following retrieval improves LLM outputs by retrieving relevant knowledge documents, enhancing the accuracy and relevance of generated content in tasks like **retrieval-augmented generation**. For instance:
>
> - If a user requires information aimed at experts, the retrieval system can pre-filter documents to surface content aligned with the expertise level, such as scientific papers or technical manuals.
> - This reduces the need for the LLM to adapt generic content to an expert audience, improving both the efficiency and the quality of the generated output.
> - Similarly, retrieving documents in specific formats (e.g., official manuals) ensures that the LLM generates text consistent with the preferred style or structure.
> By enabling retrieval systems to seamlessly align with user-defined instructions, such as audience type or document format, we aim to create a unified framework that optimizes both retrieval and subsequent LLM-based generation. This synergy is particularly valuable in applications like customer support, where domain-specific accuracy and relevance are paramount.
>
> > Document-level instructions might overlook some "detailed" features within documents
>
> In our work, document-level instructions **are not meant to replace content-level analysis but to complement it** by refining retrieval based on user-defined preferences, such as audience, format, and language. However, as pointed out, document-level instructions may overlook more detailed features within the document itself, such as specific paragraphs or sections. To address this, future enhancements could introduce more fine-grained control over retrieval, such as focusing on particular document sections (e.g., paragraphs or tables), which would allow models to retrieve content at a higher level of detail. This could be achieved by fine-tuning models with hierarchical instructions that capture both document-level attributes and finer content-specific features.
>
> In practice, however, users often prioritize broader document-level features over more granular content. In such cases, the combination of document-level retrieval models with supplementary systems, such as paragraph-level or section-based retrievers, could provide a more comprehensive approach to addressing finer granularity. By refining how instructions are aligned with document granularity, future work could make retrieval systems more robust, accurate, and tailored to user needs, offering a seamless integration of both broad and detailed search capabilities.
>
> Could you provide an example of the type of **"detailed feature"** you have in mind? This would help us better understand your concern to answer it.

---

> ### Comment · Reviewer_RSj2 · 2024-12-02
>
> Thank you for your rebuttal. I haven't had time to check the updated manuscript will now. I cannot access the version before revision but I can see you avoid using some "magic" numbers in the updated manuscript. I think my questions are well addressed in authors' response and the updated manuscript. I've increased my score accordingly.
>
> Some minor suggestions:
> 1. I like appendix B, can you somehow manage to put (some of) the analysis in appendix B into the main text?
> 2. It would be great to include the detailed annotator biographies/information in the appendix

---

> > ### Author Response · Authors · 2024-12-03
> >
> > Thank you very much for your positive feedback and for raising your score for our work. We truly appreciate your thoughtful suggestions, which will greatly enhance the clarity and comprehensiveness of the paper. We are glad you found Appendix B insightful, and as per your suggestion, we will incorporate some of the analysis from Appendix B into the main text to better highlight its relevance. Regarding the detailed annotator biographies/information, we apologize for overlooking its inclusion in the appendix and will ensure it is added as recommended to provide a more complete presentation of the annotation process. These updates will be reflected in the revised version. Thank you again for your helpful comments and support!

---

### Official Review · Reviewer_r4CY · 2024-11-07

**Soundness:** 3
**Presentation:** 3
**Contribution:** 3
**Rating:** 6
**Confidence:** 4

**Summary:**

The paper, “Beyond Content Relevance: Evaluating Instruction Following in Retrieval Models,” introduces **InfoSearch**, a benchmark that assesses how well retrieval models follow user instructions beyond simple content relevance. Traditional retrieval models have limitations in understanding and executing detailed user instructions, often focusing on keyword matching rather than instruction adherence. InfoSearch spans six key dimensions—Audience, Keyword, Format, Language, Length, and Source—representing user-specific document attributes. The study evaluates 16 retrieval models using this benchmark and introduces two novel metrics, **Weighted Instruction Sensitivity evaluation (WISe)** and **Strict Instruction Compliance Ratio (SICR)**, designed to capture the depth of instruction-following capabilities. The results reveal that current retrieval models, even those fine-tuned, often struggle with adhering to specific instructions, providing insights for advancing instruction-aware retrieval in future research.

**Strengths:**

- The writing is easy to follow and the logic is clear.
- InfoSearch covers six essential dimensions, moving beyond traditional content relevance. This multidimensional approach provides a holistic evaluation of models, accounting for varied user-specific needs that reflect real-world scenarios.
- The introduction of WISe and SICR metrics adds significant value by capturing nuanced differences in instruction-following. These metrics are specifically designed to evaluate instruction adherence, overcoming limitations in traditional metrics like nDCG or pMRR.

**Weaknesses:**

- The benchmark’s focus on instruction-following may overlook other critical aspects of retrieval quality, such as semantic understanding and contextual relevance. Overemphasizing instruction compliance may risk sidelining these equally important factors.
- The InfoSearch benchmark relies heavily on synthetic data generation, primarily using LLMs like GPT-4 to create instruction-query pairs and rephrased documents. This reliance may introduce artificial biases in the data that do not accurately reflect real-world queries or instruction styles, potentially limiting the benchmark’s validity for evaluating real-world retrieval tasks.
- While the benchmark tests models on reversely instructed queries (negative instructions), the analysis does not deeply explore how models interpret and handle negation or exclusion conditions. Understanding these nuances would be valuable for practical applications where users often specify what not to include in results.

**Questions:**

- What is the computational feasibility of using WISe and SICR metrics in large-scale production environments?

---

> ### Author Response · Authors · 2024-11-24
>
> > The benchmark’s focus on instruction-following may overlook other critical aspects of retrieval quality, such as semantic understanding and contextual relevance. Overemphasizing instruction compliance may risk sidelining these equally important factors.
>
> Existing benchmarks predominantly focus on semantic relevance and contextual understanding, which are indeed critical aspects of retrieval quality.  However, there is currently a lack of datasets and evaluations specifically designed to assess instruction-following capabilities.  The primary goal of our work is to bridge this gap by focusing on instruction compliance rather than semantic relevance alone. That said, our queries and documents are grounded in strong semantic foundations—derived from original datasets where queries and their corresponding positive documents are clearly semantically relevant.  Building on this, we have augmented the data with carefully designed instructions to enable a finer level of control and analysis.  This approach ensures that semantic understanding remains a core aspect while adding a new dimension for evaluating instruction compliance.
>
> > The InfoSearch benchmark relies heavily on synthetic data generation, primarily using LLMs like GPT-4 to create instruction-query pairs and rephrased documents. This reliance may introduce artificial biases in the data that do not accurately reflect real-world queries or instruction styles, potentially limiting the benchmark’s validity for evaluating real-world retrieval tasks.
>
> **For queries**, we curated original Q-A pairs from existing datasets, categorized as follows:
> - Audience: BioASQ, scifact
> - Keyword: MSMARCO
> - Format: Stackoverflow, Stack Exchange
> - Language: publichealth-qa
> - Length: medical_qa
> - Source: CNN-english-news
>
> The queries in MSMARCO are sourced from real user searches on Bing, while datasets like Stack Overflow and Stack Exchange also consist of questions posed by real users in real-world scenarios.
>
> **For instruction styles**, the InfoSearch benchmark generates instructions that reflect real-world user interactions, such as specifying language preferences. For example, when users require information in a specific language, the instructions explicitly direct the retrieval system to provide responses in that language. A typical instruction for the Language dimension might be: *"Please provide the answer in English."* or its reversed form, *"Please do not provide the answer in English."* These instructions mirror scenarios where users might explicitly want content in a particular language, such as when a non-English speaker requests information in their native language or when a user prefers documentation in a global language like English for consistency.
>
>
> > While the benchmark tests models on reversely instructed queries (negative instructions), the analysis does not deeply explore how models interpret and handle negation or exclusion conditions. Understanding these nuances would be valuable for practical applications where users often specify what not to include in results.
>
> We will illustrate with a specific example of e5-mistral search results.
>
> Q: how many calories are in a martini
> |              | **sentence doc** | **paragraph doc** | **article doc** |
> |:-----------------:|:----------------:|:-----------------:|:---------------:|
> | Q                 | **90.75**            | 89.42             | 84.85           |
> | Q+[sentence]      | **84.88**            | 84.01             | 78.58           |
> | Q+[paragraph]     | 84.83            | **85.50**             | 79.54           |
> | Q+[article]       | **85.76**            | 84.95             | 80.49           |
> | Q+[non-sentence]  | **86.51**            | 85.15             | 81.31           |
> | Q+[non-paragraph] | **86.57**            | 84.87             | 81.07           |
> | Q+[non-article]   | **86.82**            | 85.33             | 81.76           |
>
> Adding instructions like [sentence], [non-sentence] and so on, generally lowered the overall semantic similarity between the query and document. However, documents with originally high similarity scores maintained relatively higher scores, indicating that the model did not fully understand the instruction. The instruction merely reduced the similarity across the board without meaningfully improving relevance for the specified condition.
>
> > What is the computational feasibility of using WISe and SICR metrics in large-scale production environments?
>
> We would like to clarify that this paper focuses on offline evaluation rather than online retrieval. Specifically, we address the challenge by setting carefully designed instructions and reverse instructions for each query, ensuring that the evaluation is conducted in a controlled and consistent manner. This setup allows us to isolate and analyze the effectiveness of the methods in reranking without the variability introduced by online retrieval performance.

---

> ### Author Response · Authors · 2024-11-30
>
> Dear Reviewer,
>
> Happy Thanksgiving! Thank you for your thoughtful and detailed feedback on our paper. We believe we have addressed the key points you raised, but we are eager to know if our responses meet your expectations and if there are any remaining questions. Your continued insights would be greatly appreciated and instrumental in refining this work further.
>
> Kind regards,
>
> Author

---

> ### Author Response · Authors · 2024-12-03
>
> Dear Reviewer,
>
> As the deadline for the discussion period approaches, we would like to follow up to ensure that our latest results and clarifications address the concerns and questions you raised.
>
> Regarding your concern about the **"focus on instruction-following potentially overlooking other critical aspects of retrieval quality,"** we’d like to clarify the following:
>
> **Semantic relevance is the foundation of our benchmark.** We ensure this by using high-quality query-document pairs sourced from real-world datasets such as **MSMARCO** and **StackOverflow**. This ensures that the relevance between queries and documents is well-established. Building on these high-quality Q-D pairs, we further design and annotate instructions, with strict quality assurance through manual validation and re-annotation. These instructions enable us to **evaluate how well retrieval models comply with task-specific instructions, going beyond mere semantic relevance.** We believe that InfoSearch is the first benchmark of its kind to evaluate instruction compliance in retrieval models. This includes embedding-based models, reranking models, and zero-shot LLMs for ranking tasks, providing a more comprehensive perspective compared to benchmarks that solely assess semantic relevance.
>
> Regarding your concern on the **quality of synthetic data**, we implemented several strict validation measures. For **Model-Assisted Coarse Validation**, we used **12 retrieval models** (e.g., bge, e5-mistral, NV-Embed) to evaluate generated positive documents. **Documents were flagged if ranked below 50 or scored below 0.5 by six or more models**, then manually reviewed to confirm their validity. Hard negative documents underwent the same process to ensure contextual similarity to queries while being definitively irrelevant. For **Annotator Validation**, graduate students in computer science with proven English proficiency reviewed the annotations, achieving an inter-annotator agreement score of 92%. **Instruction Checks** were conducted for five dimensions (excluding Keyword), with manual reviews yielding 99% accuracy. For the Keyword dimension, instructions were generated by extracting keywords, with an initial 3% error rate in a sample of 20 queries. Errors were corrected iteratively using LLMs, and a re-check confirmed that all positive documents contained the specified keywords.
>
> If these updates resolve your concerns, would you mind adjusting your rating accordingly? Please let us know if there are any remaining issues we can address.
>
> Thank you for your time and feedback!
>
> Best regards,
>
> Author

---

### Official Review · Reviewer_L8BM · 2024-11-08

**Soundness:** 3
**Presentation:** 4
**Contribution:** 3
**Rating:** 8
**Confidence:** 4

**Summary:**

This paper proposes a new dataset and evaluation metrics to evaluate instruction-based information retrieval. Their dataset, InfoSearch proposes 6 different subsets of evaluation: Audience, Keyword, Format, Language, Length, and Source.

They build the dataset by collecting query and document pairs from existing IR datasets as well as Google. They then have GPT-4 extract keywords from the documents, generate an instruction via some specific style, and then re-write the document as needed. They also prompt GPT-4 to generate instruction reversals, which are the opposite. They then manually review them.

They propose two new metrics to evaluate instruction following (WISe and SICR) and show results on a wide range of models. They show that larger models do better, rerankers do better, list-wise models do better, and instruction-trained models do better. They show that in some dimensions, such as audience, are still difficult for retrievers.

**Strengths:**

I think this paper has a lot of strengths:
- A needed dataset for instruction-based IR, covering a wide range of topics, including some models still struggle on
- A wide set of models they evaluated, including larger models like GPT-4
- Interesting analysis from the metrics and various comparisons

**Weaknesses:**

Although I lean positively, there are a few weakness/area to improve

1. I am still confused slightly on the dataset creation. For example, what decided how many instructions there were per query? In Table 1 I see somewhere between 2-3 per query.  The instructions are also fairly short (20 words, which is almost standard query-length), but this is not a dealbreaker.  See the questions for these.
2. The two new introduced metrics are somewhat difficult to understand and I am not sure I fully agree with them. See the questions, from where I worked through some examples and see some ways of assigning scores that seem odd to me.
3. Although I wouldn't have expected the authors to have done this since the deadline was so soon after the model's releases, there are some models that claim to do better on this, including ICL-based models (BGE-ICL, RARe) and zero-shot models (Promptriever). It would be interesting to see their performance.

Overall, I do like the paper and would be open to increasing my score if some of these weaknesses/questions are addressed (mostly metrics based).

Update after response: I have increased my soundness score and most of my concerns have been addressed.

**Questions:**

### Re: Weakness 1, dataset construction

a. How were the number of instructions decided/created?

b. How was the number of document in the corpus decided?

c. It seems like all documents are merged together from all subsets to create the corpus, how are we sure there are not false negatives in the other sets? A qualitative analysis or GPT-based analysis would be very valuable here.

d. How were these documents gathered, just any that were from the original pairs (but then where did the remainder come from?)

### Re: Weakness 2, metrics

A core motivation from the paper of WISe is:
> Intuitively, if the ranking of Pi improves in the instructed mode (Rins < Rori) and worsens in the reversely instructed mode
(Rori < Rrev), this demonstrates the model’s effective compliance with the provided instructions

a: Why is this true? I don't necessarily understand why the query is more relevant with the instruction, weren't these gathered from already relevant query and doc pairs? So I see no specific reason why the rank should increase with the instruction as compared to no-instruction.

b: Arbitrary values: why does WISe not go to 0? Why use 20 as the number for the conditional?

c: Doesn't WISe/SICR have the same problem as p-MRR in terms of rank of 1 for the original and instructed query?

d: Working through some examples for WISe, that I found unusual:
- If the rank is 1 for both, the score is 0 correct? (From Eq 1)
- From r_{ori} of 20 to r_{ins} of 1, we have a score of 1.0. If you change r_{ins} to 2, now the score is 0.07? Perhaps I did the math wrong, but this seems like a pretty large gap
- From r_{ori} of 20 to r_{ins} of 2 we have 0.07 but from r_{ori} of 21 to r_{ins} of 1, we have 0.01 (an even lower score, even though arguably the model improved more)
- Overall, the N=20 distinction seems to introduce some weird disfluencies in the possible range of scores for this metric. Even the overall range seems somewhat odd, you can get scores from -1 to 1, but oddly distributed.

---

> ### Author Response · Authors · 2024-11-24
>
> > How were the number of instructions decided
>
> In our dataset, except for "Keyword" dimension, the other five dimensions have **fixed conditions** and we create instructions based on varied conditions(see more details in Figure 1). For example, in the "Language" dimension,  we have conditions of [English] and [Chinese], so we create instructions like "please give me  English/Chinese answers." in instructed mode. For the special "Keyword" dimension, we create about **4-6 instructions** for a core query. In the future, we will extend more dimensions and more conditions.
>
> > How was the number of document in the corpus decided
>
> In our dataset, each dimension has **100 core queries** (i.e., query with no instruction) and **about 1000 documents** (see more details in Table 1). For each core query, it has **2-5 positive documents** and multiple hard/easy negative documents.
>
> > It seems like all documents are merged together from all subsets to create the corpus, how are we sure there are not false negatives in the other sets? A qualitative analysis or GPT-based analysis would be very valuable here.
>
> Instead of merging all document subsets into a single corpus, we created a separate corpus for each dimension, as detailed in Table 1. Since some documents were rewritten using GPT, we conducted a quality check during the preparation phase to ensure reliability.
> During this test, we found some problems. 1. GPT returns invalid format; 2. GPT sometimes generates false negatives. So we carefully designed the prompt (see appendix) and manually checked the documents generated by GPT to avoid these problems.
>
> > How were these documents gathered, just any that were from the original pairs (but then where did the remainder come from?)
>
> To ensure the naturalness of the queries and reduce production costs, we integrated conditions when filtering Q-A pairs from existing datasets or web pages. For the "Keyword" dimension, the **MSMARCO** dataset was utilized, as it provides multiple positive and negative documents for each query. For the other five dimensions, existing datasets often satisfy only one condition within a dimension. To create documents meeting other conditions, we leveraged GPT to rewrite documents.
>
> > Why is this true? I don't necessarily understand why the query is more relevant with the instruction, weren't these gathered from already relevant query and doc pairs? So I see no specific reason why the rank should increase with the instruction as compared to no-instruction.
>
> For queries without instructions, all corresponding documents are treated as positive, as they are relevant to the query in a general sense. However, when instructions are added, they introduce specific requirements, making the relevance criteria more precise.
>
> Here is an example:
> - Core Q: how many calories are in a martini?
> - Q+ins1: Q + Please give me a short answer.
> - Q+ins2: Q + I'd like a paragraph explaining it.
> - Q+ins3: Q + Please provide a detailed article.
> - Doc1(sentence), Doc2(pargraph), Doc3(article)
>
> Without instructions, all documents (Doc1, Doc2, Doc3) are relevant to the query "Q."
> When "ins1" is added, Doc1 should be ranked first because the instruction specifies a short answer. Similarly, with "ins2," Doc2 (a paragraph) should take priority, and with "ins3," Doc3 (an article) should be ranked highest.
> These examples highlight how instructions refine the relevance criteria and demand that the model adapt its ranking accordingly.
>
> > why does WISe not go to 0? Why use 20 as the number for the conditional?
>
> When R_{ori} = R_{ins}, the penalty part of WISE is equal to 0, indicating perfect alignment between original and instruction-followed retrieval results. The value 20 is chosen based on the assumption that the top 20 results offer the most utility to users, balancing practical relevance and computational efficiency.
>
> > Doesn't WISe/SICR have the same problem as p-MRR in terms of rank of 1 for the original and instructed query?
>
> We address the case where R_{ori} = R_{ins} = 1 by assigning a WISe/SICR score of 1. Additionally, WISe considers both the relative distance (R_{ori} - R_{ins}) and the absolute rank (R_{ins}) in its calculation, making it more sensitive to top-ranked changes and mitigating the limitations of p-MRR.
>
> > Working through some examples for WISe, that I found unusual:
>
> - When R_{ori} = R_{ins} = 1, the condition R_{ori}< N and R_{ins} = 1 is satisfied, resulting in a WISE value of 1.
> - The rapid decay of WISE values, as observed in questions 2 to 4, is caused by the term \frac{R_{ori} - R_{ins}}{K}. To address this, we revised the formula to \frac{\sqrt{R_{ori} - R_{ins}}}{K}, which ensures smoother score transitions. Figure 5 illustrates this improvement.

---

> > ### Comment · Reviewer_L8BM · 2024-11-24
> >
> > Thanks for the response.
> >
> > > I am still confused slightly on the dataset creation.
> >
> > Your answers cleared up much of my confusion there. Part of the confusion is that, as illustrated with the example with different documents, that you have multiple instructions per query and multiple documents per query. I think some of the language could be revised to make this more clear, but it could also have been my fault. So this is resolved.
> >
> > > Working through some examples for WISe, that I found unusual:
> >
> > Thanks for pointing out the "where R_{ori} = R_{ins} = 1". I also do think that the sqrt makes it more smooth, but am not fully convinced on the metric. However, I don't think that is a reason to reject the paper as I like the other contributions and think they are valuable.
> >
> > I see Reviewer 7Ty8 doesn't like that the metric would be specific to this dataset, but it seems fine to me. Seems like it would be good at comparing systems on the dataset.

---

> ### Author Response · Authors · 2024-11-30
>
> Thank you very much for your support and positive feedback! We sincerely apologize for the confusion caused earlier. We truly appreciate your valuable input and will ensure that the details you pointed out are clarified thoroughly in the revised version.
>
> Regarding the modifications to the formula, the addition of the square root was introduced not only to ensure that the WISE score decreases monotonically as the positive rank (R_ori/R_ins) increases but also to make the score variation smoother. This adjustment addresses the issue you pointed out regarding the presence of a pretty large gap, improving the overall interpretability and consistency of the scoring mechanism.
>
> Regarding the metric design, the historical development of our evaluation approach was as follows: initially, we did not pair the instructed mode and reversely instructed mode in our evaluation framework. However, during the course of our analysis, we encountered a significant issue where the ranking of positive documents moved in the same direction across both modes, leading to false positive responses. This observation prompted us to propose a paired evaluation metric, which ensures a more accurate assessment by considering the relationship between the two modes and accounting for shifts in rankings more effectively. We believe this refinement addresses key shortcomings and enhances the reliability of the evaluation process.

---

### Meta-Review · Area_Chair_Qz6f · 2024-12-23

**Metareview:**

The paper proposes a new dataset and evaluation metrics for instruction-following in information retrieval focusing on 6 dimensions: Audience, Keyword, Format, Language, Length, and Source. The dataset is built by collecting query and document pairs from existing IR datasets as well as Google, and uses GPT-4 to generate instructions and re-write documents. The paper also proposes two new metrics to evaluate instruction following, namely, WISe and SICR that aim to better evaluate this task. Results show that larger models, rerankers, and list-wise models perform better.

*Strengths*:
-The reviewers acknowledged the importance of the dataset in measuring progress in instruction-based IR
-They found the experiments to be comprehensive
-The proposed metrics were recognized as a notable contribution
-The reviewers found the analysis from metrics and comparisons to be interesting and insightful

*Weaknesses*:
-There were several issues and places were the descriptions were confusing or there were lack of details.  Although the authors have responded to and clarified most of the issues, the fact that there were so many writing and clarity issues in the submitted version remains a concern.
-One of the reviewers mentioned that the updated manuscript violated the format, although the authors corrected that
-The motivation needed to be expanded
-One reviewer finds the dataset construction pipeline to be incremental with respect to existing datasets (see below more discussion about this).
-Some related work needed to be discussed and differentiated

This was a challenging case, involving completely opposite evaluations by reviewers and extensive discussions between reviewers and authors. To resolve the disagreements and come to a fair decision, I initiated an internal discussion between reviewers. One reviewer strongly argued that the dataset is incremental and the proposed metrics aren't well justified, deeming the contribution insufficient. In contrast, another reviewer disagreed strongly, emphasizing that by the same logic, many valuable datasets in the community could also be dismissed as incremental, arguing that the dataset provides a practical and meaningful resource for advancing instruction-following IR.

After reviewing the discussions, rebuttals, and the paper itself, I concluded that the dataset offers value to the IR and RAG communities. Instruction-following IR is an emerging area, and this resource provides a foundation for measuring and driving progress. While I acknowledge some of the opposing reviewer’s points, I believe the dataset and metrics collectively form a significant contribution.

That said, the need for numerous clarifications and revisions during the rebuttal highlights the manuscript’s lack of readiness at submission. This remains a concern and detracts from the overall state of the work at submission time.

**Additional Comments On Reviewer Discussion:**

The authors addressed most of the concerns, provided clarification and additional explanations where needed. They also provided arguments for contributions of their resource compared with existing datasets.

---

### Decision · Program_Chairs · 2025-01-22

Accept (Poster)